# Retrieval Or Holistic Understanding? DOLCE: Differentiate Our Long Context Evaluation Tasks

## Abstract

We argue that there are two major distinct capabilities in long context understanding: retrieval and holistic understanding. Understanding and further improving LLMs' long context capabilities would not be possible without knowing the tasks' focus categories. We aim to automatically identify retrieval focused and holistic understanding focused problems from suites of benchmarks and quantitatively measure the difficulty within each focus. In this paper, we present the DOLCE framework, which parameterizes each problem by $\lambda$ (*complexity*) and $k$ (*redundancy*) and assigns to one of five predefined focus categories. We propose to sample short contexts from the full context and estimate the probability an LLM solves the problem using the sampled spans. To find the $\lambda$ and $k$ for each problem, we further propose a mixture model of a non-parametric background noise component and a parametric/non-parametric hybrid oracle component, where we derive the probability functions parameterized by $\lambda$ and $k$ for both the *correct-or-wrong* (COW) scenario and the *partial-point-in-grading* (PIG) scenario. Our proposed methods can identify 0% to 67% of the problems are retrieval focused and 0% to 90% of the problems are holistic understanding focused across 44 existing long context evaluation tasks.

## 1 Introduction

Large language models (LLMs) have become capable of processing long contexts up to 10M tokens at a time (Achiam et al., 2023; Dubey et al., 2024; Anthropic, 2024; Reid et al., 2024). Model developers have also identified a large number of long context use cases and accordingly compiled existing and new long context evaluation tasks into benchmark suites to quantitatively measure LLMs' long context capabilities (Shaham et al., 2023; An et al., 2024; Dong et al., 2024; Bai et al., 2024)[1].

We argue there exist two major distinct capabilities in long context understanding: *retrieval* and *holistic understanding*. The former involves identifying a single or a few relevant pieces of information ("needle") from chunks of irrelevant content ("haystack"), while the latter assumes that a large chunk, if not all, of the content is relevant, and oftentimes even the order matters. This distinction is important since it relates to the architecture design of an efficient long context LLM. For example, divide-and-conquer approaches, such as blockwise parallel attention (Liu et al., 2024a) or parallel decoding (Li et al., 2024b), can largely improve the efficiency a Transformer model without affecting the performance on a retrieval focused task, but may put the performance of a holistic understanding task in doubt. Recurrent models (Gu et al., 2022; Bulatov et al., 2022; Gu & Dao, 2023; Poli et al., 2023; Beck et al., 2024) are believed more suited for holistic understanding, despite recent underperformance (Zhang et al., 2024a; Huang, 2024). Retrieval augmented generation (RAG) architecture is tailored for a balanced scenario. **Understanding and further improving LLMs' long context capabilities would not be possible without knowing the tasks' focus categories**, which however are sometimes unavailable. Although we may infer them from their task names (e.g. -QA or -Ret suffices often indicate retrieval) or via manual inspection, it's not reliable and time-consuming.

---

[1]We use "problem" to represent a single input or prompt, "task" to represent a group of problems that usually have a similar use case, "benchmark suite" for a group of tasks.

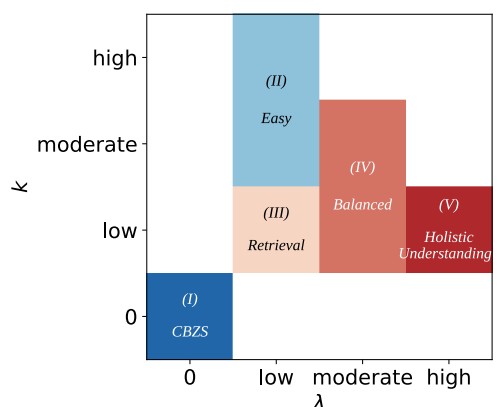

| CATEGORY | $\lambda$ | k |
|---|---|---|
| I (CBZS) | 0 | 0 |
| II (Easy) | $[1, \lambda_p]$ | $(k_p, L]$ |
| III (Retrieval) | $[1, \lambda_p]$ | $[1, k_p]$ |
| IV (Balanced) | $(\lambda_p, \lambda_q]$ | $[1, L/\lambda_p]$ |
| V (Holistic Understanding) | $(\lambda_q, L]$ | $[1, L/\lambda_q]$ |

Figure 1: Problem parameterization by $\lambda$ (complexity) and $k$ (redundancy). Category mapping is illustrated on the left and formally determined by the table on the right. $L$ represents full context, $\lambda_p$, $\lambda_q$ and $k_p$ are hyperparameters (detailed in Section 4).

We present the DOLCE (**D**ifferentiate **O**ur **L**ong **C**ontext **E**valuation Tasks) framework, which aims to automatically identify retrieval and holistic understanding focused problems from suites of benchmarks and quantitatively measure the difficulty within each focus. Intuitively, a retrieval focused problem often has a relatively short evidence span, and the difficulty depends on the number of the evidence span occurrences in the context. A holistic understanding focused problem has a longer minimum sufficient evidence span or multiple necessary spans dispersed across the context. We use two parameters $\lambda$ and $k$ to capture the span complexity and redundancy and map each area of the $\lambda$-$k$ plane to a category, as shown in Figure 1. We also define a special area for $\lambda = 0$ (no context), where the model solves the problem in a *closed-book or zero-shot* (CBZS) condition.

How can we find $\lambda$ and $k$ for each problem? One simple idea is to retrieve the spans relevant to the question and/or the ground-truth answer, and then claim that $\lambda$ is the length of the span and $k$ is the number of retrieved spans. However, this approach does not only rely on the retriever but also hardly identifies the supporting spans required in the reasoning process. In the DOLCE framework, we propose to sample short contexts from the full context and estimate the probability an LLM solves the problem using the sampled spans, which also prevents getting "lost in the middle" (Liu et al., 2024c). We note that, although an LLM can often solve short context problems better than long context problems, unlike humans or oracle models, it can still make mistakes. We therefore further propose to use a mixture model of a non-parametric background noise component and a parametric/non-parametric hybrid oracle component, making our method less sensitive to the quality of the probing model. To model the parametric oracle component, we derive the probability functions parameterized by $\lambda$ and $k$ for both the *correct-or-wrong* (COW) scenario and the *partial-point-in-grading* (PIG) scenario. We use two independently trained models: Gemini 1.5 Flash (Reid et al., 2024) and PaLM 2-S (Anil et al., 2023), to 44 tasks from three benchmark suites, and then apply the DOLCE framework to obtain their focus categories. We have identified 0% to 67% of the COW problems and 0% to 29% of the PIG problems are retrieval focused (**Category III**), and 0% to 89% of the COW problems and 0% to 90% of the PIG problems are holistic understanding focused (**Category V**). These results have helped us understand and guide development of long context capabilities of LLMs.

## 2 RELATED WORK

Long context evaluation benchmark suites have been developed, including LRA (Tay et al., 2020), ZeroSCROLLS (Shaham et al., 2023), L-Eval (An et al., 2024), LongBench (Bai et al., 2024), BAMBOO (Dong et al., 2024), LooGLE (Li et al., 2024a), Loong (Wang et al., 2024c), *LV*-Eval (Yuan et al., 2024), ∞Bench (Zhang et al., 2024b), Marathon (Zhang et al., 2024a), BABILong (Kuratov et al., 2024), Ruler (Hsieh et al., 2024), LOFT (Lee et al., 2024). Each comprises existing and/or new tasks in various domains, use cases, with contexts of different lengths and syntheticity levels. Domain and use case focused long context evaluation tasks have also been developed, including Needle-In-A-Haystack (Kamradt, 2023), LongEval (Li et al., 2023), SummHay (Laban et al., 2024), Task Haystack (Xu et al., 2024), Ada-LEval (Wang et al., 2024a), NovelQA (Wang et al., 2024b),

FLenQA (Levy et al., 2024), NoCha (Karpinska et al., 2024), RepoQA (Liu et al., 2024b). Most developers have observed performance degradation as the input context length increases. However, except in the synthetic case, they have not explicitly distinguished between two length variables: the input length ($L$) and an unknown necessary context length or complexity degree ($\lambda$).

More recent benchmarks have started to emphasize different difficulty types. Wang et al. (2024a) use the notion of "full text comprehension" for tasks whose performances "decrease sharply when the text is truncated", despite lack of detail. Li et al. (2024a) distinguish between short and long dependencies. Karpinska et al. (2024) annotate each question with a sentence, passage, or "global reasoning" scope. Wang et al. (2024c) define four categories, from "spotlight locating" to "chain of reasoning". Thus far, developers have had to manually assign categories to tasks, which can be unreliable and costly. We believe that difficulty should be a continuous spectrum, rather than categorical, and we need a quantitative approach to automatically assign categories. Along this line, Qian et al. (2024) propose LC-Boost, which iteratively interacts with an LLM agent to find "minimal necessary context". They adopt a simplified assumption of ours (with no $k$), but the quality highly relies on the retriever and the LLM. Since their goal is to solve the tasks, rather than analyze the tasks, they do not report the minimal necessary context lengths. Most relevant to ours, Goldman et al. (2024) coincidentally propose two difficulty dimensions: scope and diffusion, in a position paper. Our DOLCE framework not only formally defines $\lambda$ and $k$ and derive probability functions under two different assumptions, but also quantitatively estimates $\lambda$ and $k$. To the best of our knowledge, our paper is the first to study the problem of automatic categorization of long context tasks.

## 3  DOLCE: DISTINGUISH OUR LONG CONTEXT EVALUATION TASKS

The DOLCE framework consists of two major steps: sampling & observation and parameter estimation. In the first step, we use a probing model to observe responses given sampled short contexts, which are then evaluated. We describe this step in Section 3.1. In the second step, we attempt to find $\lambda$ and $k$ that maximize the likelihood of the observed evaluation outcomes. We use a mixture model assumption that smooths out the model noise. The modeling process slightly differs between the COW and PIG scenarios, which are defined and discussed in detail in Sections 3.2 and 3.3.

### 3.1  SAMPLING & OBSERVATION

For a given problem, we first chunk its context into $L$ *units*, where $L$ is also referred to as the length of the context. We choose sentences as units in most cases, but also consider other granularities when explicit structures are available. We define a *span* as a sequence of contiguous units. We randomly sample an *observation span* of length $C$ from the full context, and then observe an evaluation outcome x. We may also iterate over all the possible spans (by shifting one unit at a time) when budget allows.

When using a binary evaluation metric, e.g. accuracy, the random variable x can only take two or three values: "1" meaning fully correct, "0" meaning totally incorrect, and optionally "IDK" (or $\emptyset$ for brevity) when not enough information is provided, if instructed. We refer to this as the *correct-or-wrong* (COW) scenario. When using a continuous evaluation metric, e.g. F-1 or ROUGE, the interpretation of x may be ambiguous. Some problems lack a comprehensive list of answer variants and instead employ F-1 for fuzzy matching. In this case, we should find a threshold to binarize x and treat it as the COW case. Other problems that expect multi-aspect answers use continuous metrics to allow partial points. In this case, we need an alternative *partial-point-in-grading* (PIG) scenario to directly incorporate the raw continuous outcome x. We see in the following subsections that these two scenarios will lead to different assumptions and probability functions. We use Hartigans' Dip Test (Hartigan & Hartigan, 1985) based on the collective observed outcomes to classify each problem into either COW or PIG scenario, since both scenarios can co-exist in the same task. In particular, we bucketize the scores into bins of equal width of $0.1$, and assign COW to a problem if the p-value is below 0.5, i.e. a multi-modal score distribution, and PIG otherwise.

Once we make multiple observations of different lengths and collect evaluation outcomes, we may guess the length of a minimum sufficient span that can answer the question, which can be one definition for $\lambda$. We consider an example of a COW scenario in Table 1. An optimistic person would say $\lambda = 1$ because that's when the model starts to output the correct answer ($P(x = 1) > 0$), and a pessimistic person would say $\lambda = 20$ because that's when the model never produces an incorrect

Table 1: Example outcomes from multiple observations for a single problem.

| OBSERVATION LENGTH | 0 | 1 | 2 | 5 | 10 | 20 | 50 | 100 | FULL |
|---|---|---|---|---|---|---|---|---|---|
| $P(\mathrm{x} = 1)$ | 0.00 | 0.21 | 0.18 | 0.20 | 0.25 | 0.29 | 0.41 | 1.00 | 1.00 |
| $P(\mathrm{x} = 0)$ | 0.00 | 0.12 | 0.12 | 0.11 | 0.05 | 0.00 | 0.00 | 0.00 | 0.00 |
| $P(\mathrm{x} = \emptyset)$ | 1.00 | 0.66 | 0.70 | 0.69 | 0.70 | 0.71 | 0.59 | 0.00 | 0.00 |

answer ($P(\mathrm{x} = 0) = 0$). In fact, the same model may make a lucky guess sometimes, and produce an incorrect answer other times.

### 3.2 Correct-Or-Wrong (COW) Scenario

**Mixture of Noise & Oracle Components.** We assume both a background noise component $\mathcal{N}$ and an oracle component $\mathcal{O}$ reside inside the probing model, and they jointly produce the final outcome $\mathrm{x}_{ij}$. The probability $P(\mathrm{x}_{ij} = x_{ij})$ is a mixture of two generation processes. We use $i$ to represent a problem and $j$ for a sampled span for problem $i$.

$$P(\mathrm{x}_{ij} = x_{ij}) = \sum_{z \in \{\mathcal{N}, \mathcal{O}\}} P(\mathrm{x}_{ij} = x_{ij}|\mathrm{z}_{ij} = z)P(\mathrm{z}_{ij} = z) \tag{1}$$

where $\mathrm{z}_{ij}$ is a latent random variable, taking either $\mathcal{N}$ (noise) or $\mathcal{O}$ (oracle).

**Background Noise Component.** We use the term background noise to describe the process that the model outputs the answer without referring to or understanding the given context, which does not imply that the answer must be wrong. In fact, there are several scenarios that a background noise component can produce correct answers. First, a dummy model can guess the correct answer with a probability of $1/4$ for a four-choice question. Also, a model can correctly answer some questions when zero context is provided (i.e. a closed-book test setup), possibly due to the fact that it has seen and memorized the hidden contexts during training, which is also noted by prior benchmark developers (Dong et al., 2024; Li et al., 2024a; Wang et al., 2024c).

We make a non-parametric assumption that the background noise has three outcomes with constant but unknown probabilities, i.e., $P(\mathrm{x}_{ij} = x_{ij}|\mathrm{z}_{ij} = \mathcal{N})$ is given by

$$P(\mathrm{x}_{ij} = x_{ij}|\mathrm{z}_{ij} = \mathcal{N}) = p_{\mathcal{N},1}{}^{[\![x_{ij}=1]\!]} \; p_{\mathcal{N},0}{}^{[\![x_{ij}=0]\!]} \; p_{\mathcal{N},\emptyset}{}^{[\![x_{ij}=\emptyset]\!]} \tag{2}$$

where $p_{\mathcal{N},1}, p_{\mathcal{N},0}, p_{\mathcal{N},\emptyset}$ are the only three parameters.

**Oracle Component.** We assume there exists a span of length $\lambda$ that contains all the necessary pieces of information for the oracle model to confidently answer the question. In many long context problems, such length-$\lambda$ ground-truth spans may appear multiple times in different parts of the input context (e.g. concatenated search result pages for a given query). The oracle model only needs to find any of them. Since we want to find the shortest ground-truth span, we further assume that the oracle model cannot answer this question if it only sees a partial span.

**Assumption 1 (COW: $k$-repeated length-$\lambda$ sufficient spans)** *There exist $k$ non-overlapping ground-truth spans, each with a length of $\lambda$ units ($k\lambda \leq L$). An observation of length $C$ can answer this question if and only if the observation span completely covers one of the $k$ ground-truth spans.*

We derive combinatorially the probability $\pi(\lambda, k; L, C)$ that the observation span covers the ground truth span. The formula and the derivation are provided in Appendix A.1 and an example distribution illustration is given in Appendix A.2. Under this assumption, the oracle model should observe "1" with a probability equal to $\pi$, and $\emptyset$ with a probability equal to $1 - \pi$. In most cases, the oracle should never make mistakes, i.e. observe "0". Then, the probability mass function (pmf) $P_{\mathrm{par}}(\mathrm{x}_{ij} = x_{ij}|\mathrm{z}_{ij} = \mathcal{O}; \lambda_i, k_i)$ can be written as:

$$P_{\mathrm{par}}(\mathrm{x}_{ij} = x_{ij}|\mathrm{z}_{ij} = \mathcal{O}; \lambda_i, k_i) = \pi(\lambda_i, k_i; L_i, C_{ij})^{[\![x_{ij}=1]\!]} \; 0^{[\![x_{ij}=0]\!]} \; (1 - \pi(\lambda_i, k_i; L_i, C_{ij}))^{[\![x_{ij}=\emptyset]\!]} \tag{3}$$

**Hybrid Oracle Component.** We note that even the oracle model can make "correct" mistake. Consider the question from the TopicRet task in the L-Eval suite: "What is the first topic we

discussed?" When the oracle model is presented a span that only discusses the second topic, it may output this topic since it is the first topic it sees. This type of mistake is different from a mistake caused by the background noise component, where the latter does not understand what "first" means and/or what "topic" means and outputs a random word. To accommodate this scenario, we further propose a hybrid assumption that combines the parametric assumption in Eq 3 and a non-parametric assumption similar to the background noise assumption. Specifically, the non-parametric assumption first applies when the observation length $C < \lambda$, in which stage the output remains "chaotic" and "0" is a valid outcome. $P_{\text{nonpar}}(\mathbf{x}_{ij} = x_{ij} | \mathbf{z}_{ij} = \mathcal{O})$ is given by

$$P_{\text{nonpar}}(\mathbf{x}_{ij} = x_{ij} | \mathbf{z}_{ij} = \mathcal{O}) = p_{\mathcal{O},1}^{[\![x_{ij}=1]\!]} \, p_{\mathcal{O},0}^{[\![x_{ij}=0]\!]} \, p_{\mathcal{O},\emptyset}^{[\![x_{ij}=\emptyset]\!]}$$

And then, the parametric assumption applies when the observation length $C \geq \lambda$. The hybrid assumption defines $P(\mathbf{x}_{ij} = x_{ij} | \mathbf{z}_{ij} = \mathcal{O}; \lambda_i, k_i)$ as follow:

$$P(\mathbf{x}_{ij} | \mathbf{z}_{ij} = \mathcal{O}; \lambda_i, k_i) = P_{\text{nonpar}}(\mathbf{x}_{ij} | \mathbf{z}_{ij} = \mathcal{O})^{[\![C_{ij} < \lambda_i]\!]} P_{\text{par}}(\mathbf{x}_{ij} | \mathbf{z}_{ij} = \mathcal{O}; \lambda_i, k_i)^{[\![C_{ij} \geq \lambda_i]\!]} \quad (4)$$

### 3.3 Partial-Point-In-Grading (PIG) Scenario

**Mixture of Noise & Oracle Components.** Similar to the COW scenario, we assume the final outcome $\mathbf{x}_{ij} = s_{ij}$ is a result of a mixture of noise background component and oracle component. Different from the COW scenario, the PIG scenario assumes the mixture happens at the sub-unit level, e.g. unigram, bigram, etc., depending on the metric (ROUGE-1, 2, etc.). We use a random variable $\mathbf{y}_{ijl}$ to represent whether the output from $j$-th observation for the $i$-th problem also contains the $l$-th sub-unit in the ground-truth answer. While $\mathbf{x}_{ij}$ is a continuous variable, $\mathbf{y}_{ijl}$ is a binary variable. In particular, it is considered a hit ("1") if the sub-unit is also identified in the prediction, and a miss ($\emptyset$) otherwise. Similar to Eq. 1, we can write the sub-unit level mixture for $P(\mathbf{y}_{ijl})$.

$$P(\mathbf{y}_{ijl} = y_{ijl}) = \sum_{z \in \{\mathcal{N}, \mathcal{O}\}} P(\mathbf{y}_{ijl} = y_{ijl} | \mathbf{z}_{ijl} = z) P(\mathbf{z}_{ijl} = z)$$

We note that the final outcome $s_{ij}$ is in fact $P(\mathbf{y}_{ijl})$. We can also use $s_{\mathcal{N}}$ and $s_{\mathcal{N}}$ to represent the component level outcomes, i.e. $s_{ij} = P(\mathbf{y}_{ijl})$, $s_{\mathcal{O},ij} = P(\mathbf{y}_{ijl} | \mathbf{z}_{ijl} = \mathcal{O})$, and $s_{\mathcal{N},ij} = P(\mathbf{y}_{ijl} | \mathbf{z}_{ijl} = \mathcal{N})$. Then, we can rewrite the sub-unit level mixture as

$$s_{ij} = s_{\mathcal{O},ij} P(\mathbf{z}_{ij} = \mathcal{O}) + s_{\mathcal{N},ij} P(\mathbf{z}_{ij} = \mathcal{N})$$

Intuitively, the final outcome $s_{ij}$ lies on the line segment with endpoints at $s_{\mathcal{O},ij}$ and $s_{\mathcal{N},ij}$. The distances to the two endpoints are inversely proportional to the respective priors. We also have

$$p(\mathbf{x}_{ij} = s_{ij}) = \sum_{z \in \{\mathcal{N}, \mathcal{O}\}} p(\mathbf{x}_{ij} = s_{z,ij} | \mathbf{z}_{ij} = z) P(\mathbf{z}_{ij} = z) \quad (5)$$

**Background Noise Component.** We simply assume $p(\mathbf{x}_{ij} = s_{\mathcal{N},ij} | \mathbf{z}_{ij} = \mathcal{N}) = 1$, i.e. a uniform distribution, meaning that we have no preference (or prior) over the underlying probability distribution $s_{\mathcal{N},ij}$. We can also consider other prior, i.e. Gaussian or beta.

**Oracle Component.** We assume there exist $\lambda$ length-1 aspects distributed across the context, each repeating $k$ times. The partial point the oracle model will get is proportional to the number of aspects the observation span covers.

**Assumption 2 (PIG: $k$-repeated $\lambda$ length-1 aspects)** *There are $\lambda$ span groups, each having $k$ unit spans. All $k\lambda$ spans do not overlap and are uniformly distributed. An observation span of length $C$ covers a span group if it covers at least one of $k$ members of the group. A partial point $s$ is awarded if the observation span covers exactly $s\lambda$ span groups.*

We can also derive combinatorially the discrete probability $\tilde{\rho}(s, \lambda, k; L, C)$ that a partial point $s$ is awarded, where $s$ must be a multiple of $1/\lambda$. The formula and derivation are given in Appendix B.1. We further transform it into a continuous probability function $\rho(s, \lambda, k; L, C)$ for an arbitrary outcome $s \in [0, 1]$. We describe the detail in Appendix B.2 and illustrate an example distribution of $\rho$ in Appendix B.3, where we also compare and explain the difference between the $\pi$ and $\rho$ derived from the two assumptions. We define $p(\mathbf{x}_{ij} = s_{\mathcal{O},ij} | \mathbf{z}_{ij} = \mathcal{O}; \lambda_i, k_i)$ as

$$p(\mathbf{x}_{ij} = s_{\mathcal{O},ij} | \mathbf{z}_{ij} = \mathcal{O}; \lambda_i, k_i) = \rho(s_{\mathcal{O},ij}, \lambda_i, k_i; L_i, C_{ij}) \quad (6)$$

### 3.4 MAXIMUM LIKELIHOOD ESTIMATION OF $\lambda, k$

The goal for maximum likelihood estimation (MLE) is to find $\lambda$ and $k$ that best describe the data $\{x_{ij}\}_{ij}$ by optimizing the joint distribution $\{p(x_{ij})\}_{ij}$ using Eq 1 or 5. We use the expectation-maximization technique to solve our mixture problem. We use the standard E-step to compute the posterior probabilities $q(z_{ij} = z_{ij}|x_{ij} = x_{ij})$ or $q(z_{ij} = z_{ij}|y_{ijl} = y_{ijl})$, and the M-step to compute the parameters, including $p_{z,x}$ ($z = \mathcal{O}, \mathcal{N}$ and $x = 0, 1, \emptyset$), membership priors, $\lambda$ and $k$.

Optimizing the parameters $\lambda$ and $k$ in these combinatorial probability functions is difficult. Also, since all $C$, $\lambda$ and $k$ can vary from 0 or 1 to $L$, we cannot easily approximate these functions using the asymptotic techniques. Fortunately, our goal is not to find the exact optimal parameters, instead we can provide a small set of $\lambda$ and $k$ candidates, using exponential intervals $(0, 1, 2, 5, 10, 20, \ldots)$, and find the maximum probability only among these combinations[2]. We provide pseudocodes of our EM-based MLE algorithm for the two scenarios in Appendix C. Finally, we use the assignment criteria described in Figure 1 to assign a category label to each problem.

## 4 PREPROCESSING & SETUPS

We identified three most cited new benchmark suites at the time of our preparation: L-Eval (An et al., 2024), BAMBOO (Dong et al., 2024), and LongBench (Bai et al., 2024), and collected a total of 44 tasks, which also include most tasks in ZeroSCROLLS (Shaham et al., 2023). We understand that most contexts used in these suites have much fewer than 100K tokens, which are considered only "moderately long" by the current standard. Yet, we found that no task in the COW scenario has all **Category V** problems, i.e. $\lambda < L$. We first apply the task specific preprocessing steps (described in Appendix D), and expand the prompts with IDK instructions (exemplified in Appendix E). Next, we find the most appropriate unit granularity and determine the observation lengths. We show an example of the study in Appendix F and report the sampling and observation specs in Appendix G.

We primarily use the Gemini 1.5 Flash model (Reid et al., 2024), unless otherwise noted. We follow the sampling and observation procedure described in Section 3.1. The tasks are evaluated using accuracy, F-1, ROUGE-L, or EditSim, against the provided ground-truth answers. We conduct the Hartigans' Dip Test to the 29 tasks evaluated using F-1, ROUGE, or EditSim, and we found that 4 tasks have COW only problems, 10 tasks have PIG only problems, and the 15 tasks have a combination of COW and PIG problems. We present the Dip Test results in Appendix H. During MLE, we optimize the likelihood function in the COW scenario (Eq. 1) for the accuracy-evaluated problems as well as Dip Test identified COW subsets using a threshold of 0.5, and the likelihood function in the PIG scenario (Eq. 5) for the Dip Test identified PIG subsets.

In both scenarios, $p(x|z = \mathcal{N})$ is shared across all problems of the same task, and $P(z)$ is shared across all samples of the same problem. In the COW scenario, $P_{\text{nonpar}}(x|z = \mathcal{O})$ is shared across all samples of the same problem, computed using only the outcomes when $C < \lambda$. In the PIG scenario, $P_{\text{nonpar}}(y|z = \mathcal{O})$ is shared across all samples with the same observation length. During parameter inference, we try $\lambda_i$ and $k_i$ from $\{C_{ij}\}_j \cup \{0, \max_j C_{ij} + 1, L_i\}$, where $\max_j C_{ij} + 1$ can help identify **Category V** problems in the PIG scenario, since the PIG assumption always expects $s_{ij} = C_{ij}/L$ when $\lambda = L$, which happens rarely. We run the EM algorithm for 10 steps in the COW scenario and 5 steps in the PIG scenario. We set thresholds $\lambda_p$ and $k_p$ as the first tertile among the $N$ exponential candidates (excluding $0, \max_j C_{ij} + 1, L$), i.e. $p = \lfloor N/3 \rfloor$. We use $\lambda_q = \max_j C_{ij}$. Four tasks exist in two suites, in which cases we use the smaller threshold for both tasks.

## 5 RESULTS

We note that, while $\lambda$ and $k$ are chosen objectively via MLE, the category assignment may be subjective, due to our choices of thresholds. These categories are nonetheless a reasonable simplification.

We report our main results in Figure 2, where tasks or task subsets are sorted by the total percentage of **Categories III** to **V** among their COW or PIG peers. In Figures 9 and 10 in Appendix I, we further sort the tasks by the percentage of retrieval focus (**Category III**) and holistic understanding focus

---

[2]We know these functions are not convex, but we suspect that they are unimodal. If so, we may also improve the optimal solution search process.

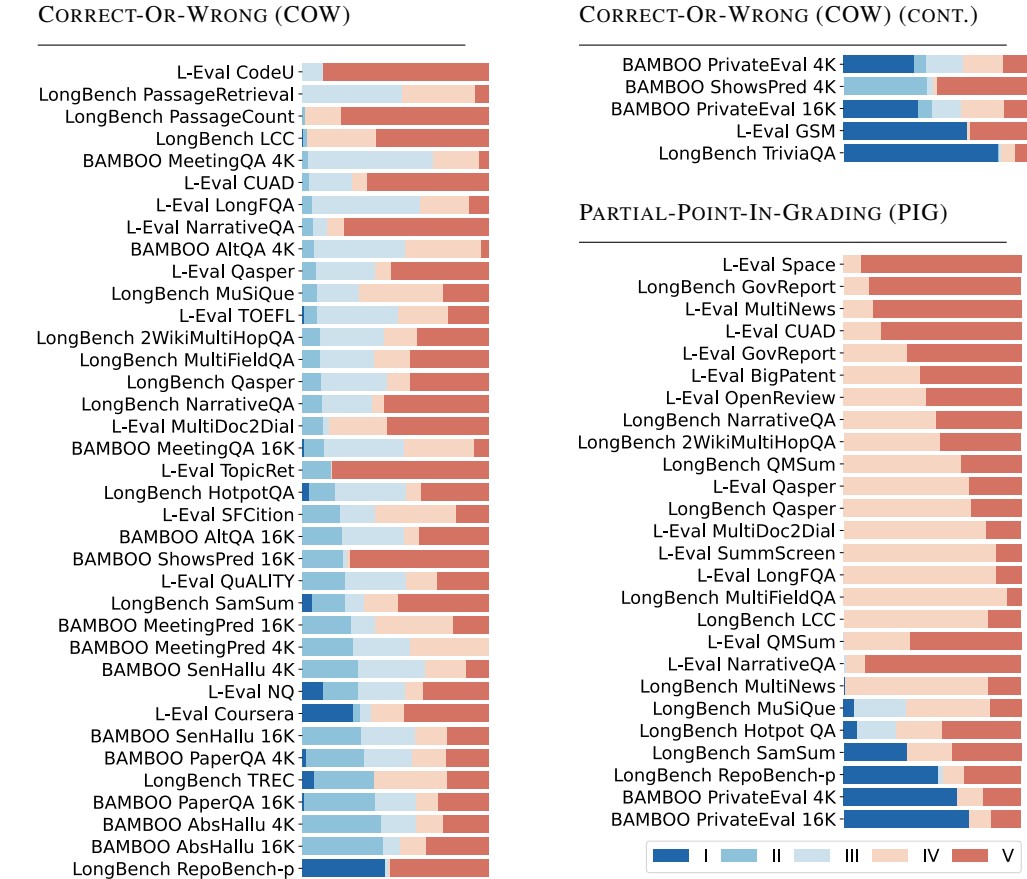

Figure 2: Task focus categories. Tasks are sorted by the total percentage of Categories III to V.

(**Category V**). In sum, we found that 0% to 67% of the COW problems and 0% to 29% of the PIG problems are retrieval focused (**Category III**), and 0% to 89% of the COW problems and 8% to 90% of the PIG problems are holistic understanding focused (**Category V**).

## 5.1 CORRECT-OR-WRONG (COW) SCENARIO RESULTS

First, we see that a few COW tasks/subsets, e.g. TriviaQA and GSM, have a large percentage of the questions that can be solved without the provided context (**Category I**), suggesting that the model may have already seen and memorized the relevant contexts (and possibly alongside the questions and answers) during training, or the questions have contained all the necessary relevant information. Second, we see that the binary classification tasks (e.g. SenHallu and AbsHallu) and few-class classification tasks (e.g. TREC, ShowsPred, where the latter often has very few candidates) tend to have more easy questions (**Category II**), especially we found the model can more often answer "yes" correctly in the SenHallu and AbsHallu tasks, even with short contexts, possibly based on its own internal knowledge, in the same way as **Category I**, but refuses to answer when no context is given.

In Figure 9(a), we sort the tasks from the most retrieval focused to the least retrieval focused (**Category III**). MeetingQA, which contains a number of factoid questions (e.g. "What additional funding has been committed by the Welsh Government to support people arriving from Ukraine?"), is ranked at the top. In fact, information seeking tasks, including PassageRetrieval and most QA tasks are ranked higher in the list. In contrast, PassageCount, GSM, and TopicRet, and LCC's COW subset are the least retrieval focused tasks. Tasks that require more holistic understanding (**Category V**) are mostly those that challenge retrieval capability less, which include coding problems (LCC's COW subset and CodeU), counting problems (PassageCount), and questions that involve ordinals (TopicRet), as shown in Figure 9(b).

Table 2: The most representative problem from each focus category for QuALITY and LongFQA.

| CAT | QuALITY | LongFQA |
|---|---|---|
| II | Why might one not want to live in the universe in which this story takes place? ($\lambda = 1$, $k = 100$) | What are some key accomplishments of FS KKR Capital Corp. in 2018 as mentioned in the call? ($\lambda = 5$, $k = 10$) |
| III | Why does the text mean when it says that Korvin was "unconscious" at the time of his lessons in the local language? ($\lambda = 1$, $k = 1$) | What were the consolidated revenue and the revenue growth for the Surgical Product segment over last year? ($\lambda = 2$, $k = 1$) |
| IV | Why would Tom Dorr frame Asa Graybar for stealing the Slider egg? ($\lambda = 50$, $k = 1$) | What impact did the introduction of the Valved Tearaway and Pediatric Microslide Introducer products have on the company's market position and potential future sales? ($\lambda = 20$, $k = 1$) |
| V | How many sentences does this story have approximately? ($\lambda = L$, $k = 1$) | Does JLL have greater market share in U.S. leasing than in Capital Markets? ($\lambda = L$, $k = 1$) |

We also found that structural format (e.g. passages for PassageRetrieval and paragraphs for MeetingQA) could help shift the problem focus from more holistic understanding to more retrieval in the spectrum. Passages and paragraphs are often self-contained, as opposed to algorithmically identified sentences or lines of code. Relevant content can be found in fewer units as a result.

## 5.2 PARTIAL-POINT-IN-GRADING (PIG) SCENARIO RESULTS

For the PIG tasks and subsets, PrivateEval and RepoBench-p's PIG subsets are identified as CBZS (**Category I**). We suspect that the model may have seen their contexts or tasks during training. HotpotQA and MuSiQue's PIG subsets also have a large percentage of easy (**Category II**) problems.

We see, from Figure 10, that only two QA tasks, MuSiQue and HotpotQA's PIG subsets, still have a substantial percentage of retrieval focused problems (**Category III**). The majority of the PIG tasks and subsets consists of balanced (**Category IV**) and holistic understanding focused (**Category V**) problems. Among these tasks, we found that tasks that require first retrieval, then reasoning and summarization, e.g. LongFQA, MultiFeildQA, and other QA tasks, are classified as balanced (**Category IV**). Document summarization and long sequence generation tasks, e.g. GovReport, SPACE, and OpenReview, tend to be considered holistic understanding focused (**Category V**).

## 5.3 EXAMPLES: QUALITY & LONGFQA

In this section, we look into two tasks: QuALITY and LongFQA. Both tasks have a blend of problems from **Categories II** to **V** under the COW assumption. We show the category assignment of the most representative problems in Table 2 and present the full answer, the raw outcomes, and intermediate parameters in Appendix J. The most representative problems for **Categories II** to **V** are defined as min $\lambda$-then-max $k$, min $\lambda$-then-min $k$, max $\lambda$-then-max $k$, and max $\lambda$-then-min $k$ respectively.

We try to "speculate" the rationale behind the assignments. **Category II** is assigned to the first QuALITY problem ("Why might ..."), since the correct answer "Survival itself is difficult" is also true in our universe that the model is exposed to. The same **Category II** is assigned to the first LongFQA problem ("What are ..."), since the answer ("receiving shareholder approval ...") is repeated multiple times in the speech. The most representative **Category III** problems for both tasks seem to ask for very specific details of a fact mentioned in the context. It is even more obvious that the LongFQA problem ("What were ...") is a factoid question. The **Category IV** problem ("What impact ...") for the LongFQA task is a set question, requiring collecting multiple facets from a longer span. The **Category V** problem for the QuALITY task ("How many ...") requires to count the number of sentences, exhibiting a clear holistic intent. Despite the yes/no form of the **Category V** problem of the LongFQA task ("Does JLL ..."), the provided transcript does not disclose the firm's market share in either division at all. However, a human may guess "yes" from some clues, e.g. the speaker emphasized leasing more than capital market ("leasing and capital market" thrice vs. never "capital market and leasing"), and hinted that leasing was a more established business than capital market.

## 6 FURTHER ANALYSIS & DISCUSSIONS

Modeling decisions may affect the estimation of parameters. We define a few metrics to quantify the difference between two sets of parameters estimated from the reference (**ref**) and the **test** setups, including **Relative Change ($\delta$)** of $\lambda$ and $k$, defined as $\delta(\lambda) = \frac{|\lambda_{\text{ref}} - \lambda_{\text{test}}|}{\max(\lambda_{\text{ref}}, \lambda_{\text{test}})}$ and $\delta(k) = \frac{|k_{\text{ref}} - k_{\text{test}}|}{\max(k_{\text{ref}}, k_{\text{test}})}$, **Spearman's rank correlation coefficient ($\rho$)** of $\lambda$ and $k$, and **KL Divergence** of $p(\text{x}|\text{z} = \mathcal{N})$. We summarize our findings in this section and provide details in Appendix K.

**Sampling Strategies.** We implement a few heuristics based sampling strategies and found that the take-every strategy (i.e. shifting the observation window by a fixed number of units) generally works well. In the case of L-Eval SFcition, using take-every-5 strategy (i.e. reducing the total required resource by 80%), we can still obtain $\rho(\lambda)$ of 0.93, $\rho(k)$ of 0.99, and KL Divergence of $P(\text{x}|\text{z} = \mathcal{N})$ of $3.7 \times 10^{-5}$. We provide more details in Appendix K.1.

**Unit Granularities.** We use different unit granularities for seven tasks. We see that in the COW scenario, the rankings of both $\lambda$ and $k$ are preserved between different unit granularities, with $\rho(\lambda)$ between 0.54 and 0.80 and $\rho(\lambda) \geq 0.50$, i.e. strong correlation. The ranking of $\lambda$ is sometimes less preserved in the PIG scenario, with $\rho(\lambda)$ between 0.35 and 0.84 across the tasks, i.e. moderate to strong correlation, while the ranking of $k$ is also well preserved with $\rho(k) \geq 0.64$. The background noise distribution estimation is mostly preserved as well, with the KL divergence $\leq 0.18$ across all tasks. We provide more details and explanations for the minor discordance in Appendix K.2.

**Probing Models: Gemini 1.5 Flash vs. PaLM 2-S.** We apply the PaLM 2-S model to the same COW and PIG splits determined by the Hartigans' Dip Test results using the Gemini 1.5 Flash model scores. We compare $\lambda$ and $k$ estimated by the two models across all tasks and further ignore the problems assigned to **Category I** by either model when computing $\delta$ and $\rho$. The median $\delta(\lambda)$ and $\delta(k)$ are 0.30 and 0.16, and the median $\rho(\lambda)$ and $\rho(k)$ are 0.43 and 0.41, which fall into the moderate correlation category. We provide more details and our thoughts on the disagreement in Appendix K.3.

**Binarization Thresholds In Adapting COW Assumption For Continuous Scores.** We compare between the default binarization threshold (0.5) with 0, 0.25, 0.75, and 1 for the problems in the tasks identified as the COW scenario by the Hartigans' Dip Test. We found that, as we increase the threshold, the category assignment either does not change or shifts from **Category II** or **III** towards **Category IV** or **V**. When the threshold is changed from 0.5 to 0.25, 0.75, or 1, $\rho(\lambda)$ and $\rho(k)$ are above 0.48 and 0.41 across all but two tasks. When the threshold is changed to 0, both $\rho(\lambda)$ and $\rho(k)$ decrease, suggesting the threshold must be greater than 0. We give more details in Appendix K.4.

**Same Tasks From Different Benchmark Suites.** Four tasks exist in both L-Eval and LongBench suites. We found that only the Qasper task has similar category distributions, and the L-Eval versions of MultiNews and NarrativeQA have more holistic understanding "flavor" than the LongBench versions, but the L-Eval version of GovReport has less holistic understanding than the LongBench version. The discrepancy can be explained by the different problem selection schemes, leading to different median context lengths and difficulty levels. We give more details in Appendix K.5.

**Application In Model Development: KV Cache Update Schedule.** This work is motivated by our and others' observations that different long context LLM architectures may behave differently for different categories of long context tasks. In Appendix K.6, we present a case study on the least recently attended (LRA) (Yang & Hua, 2024), an efficient KV cache update schedule for long context LLMs. We found that if we want to utilize LRA to improve the efficiency of a long context application, we need to understand its focus category and adjust the input format accordingly.

## 7 CONCLUSION & FUTURE WORK

In this paper, we introduce two parameters $\lambda$ and $k$ to quantitatively measure the difficulty along the two dimensions: complexity and redundancy. Then, we propose the DOLCE framework that leverages a mixture model to estimate these parameters. Our proposed methods can identify 0% to 67% of the problems are retrieval focused and 0% to 90% of the problems are holistic understanding focused across the tasks and scenario subsets. We also acknowledge that our paper has some limitations, which we summarize in Appendix L. Practically, we plan to apply our framework to more recent longer context tasks to help categorize their focuses.

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

# A DETAILS OF $\pi$ IN THE COW SCENARIO ($k$-REPEATED LENGTH-$\lambda$ SUFFICIENT SPANS)

## A.1 FORMULA & DERIVATION

We omit the subscript $ij$ in this subsection.

The cover probability for the COW scenario ($k$-repeated length-$\lambda$ spans) $\pi(\lambda, k; L, C)$ is given by:

$$\pi(\lambda, k; L, C) = 1 - \frac{2\binom{w+\lambda}{k+1} + \frac{k-1}{k+1}(2k\lambda + 2\lambda + w - 2k - ku - 1)\binom{w+u}{k}}{\binom{w+C}{k}(L - C + 1)}$$

where

$$w = L - C - k\lambda + k \quad \text{and} \quad u = \min(C, 2\lambda - 2)$$

We set $\pi = 0$ when $k < 1$, $k\lambda > L$ or $C < \lambda$. There could be multiple ways to derive this combinatorial expression. We provide one derivation below.

**Derivation.** There are a total of three scenarios that the observation does not cover a single valid ground truth span.

**Scenario 1: The observation and the ground truth span do not overlap.**

The number of times this happens can be calculated via the star-and-bar process. The first step involves inserting ground-truth spans and the second step involves inserting observation span. The number of combinations for this scenario is then given by

$$\binom{L - C - k\lambda + k}{k}(L - C - k\lambda + k) = \binom{w}{k}w = (k+1)\binom{w+1}{k+1}$$

where $w = L - C - k\lambda + k$ is the sequence length before any ground-truth span or observation span is inserted.

**Scenario 2: The observation and a ground truth span overlap by $x$ positions on one side.**

This scenario describes a case where one side of the observation span partially covers a ground truth span (by $x$). But since $x < \lambda$, this is still a failure case. Similar to Scenario 1, the number of combinations in this scenario can also be derived via the star-and-bar process, as follows:

$$2\binom{L - (C + \lambda - x) - (k-1)\lambda + k - 1}{k - 1}(L - (C + \lambda - x) - (k-1)\lambda + k)$$

$$= 2\binom{L - C - k\lambda + k + x}{k}k = 2k\binom{w + x}{k}$$

We have a 2-multiplier since the partial overlapping can happen at either side of the observation. Since $x$ can have a range from 1 to $\lambda - 1$, the total number of combinations is given by

$$\sum_{x=1}^{\lambda-1} 2k\binom{w + x}{k} = 2k \sum_{x=w}^{w+\lambda-1} \binom{x}{k} = 2k\left(\sum_{x=0}^{w+\lambda-1}\binom{x}{k} - \sum_{x=0}^{w}\binom{x}{k}\right) = 2k\binom{w+\lambda}{k+1} - 2k\binom{w+1}{k+1}$$

**Scenario 3: The observation and a ground truth span overlap by $x$ positions from both sides.**

Now since that the overlap must happen on both sides of the observation, we should have $x - 1 \leq \lambda - 1$ and $x - (\lambda - 1) \geq 1$. Hence, the total number of possible left and right overlapping cases is

$$\min(\lambda - 1, x - 1) - \max(1, x - (\lambda - 1)) + 1 = \min(\lambda, x) - \max(\lambda, x) - 1 + \lambda = -|x - \lambda| + \lambda - 1$$

Similar to Scenarios 1 and 2, the total number of combinations in this scenario when there are a total of $x$ overlapping positions from both sides is given by

$$(-|x - \lambda| + \lambda - 1)\binom{L - (C + 2\lambda - x) - (k-2)\lambda + k - 2}{k-2}(L - (C + 2\lambda - x) - (k-2)\lambda + k - 1)$$

$$= (-|x - \lambda| + \lambda - 1)\binom{L - C - k\lambda + k + x - 1}{k-1}(k-1)$$

$$= (-|x - \lambda| + \lambda - 1)(k-1)\binom{w + x - 1}{k-1}$$

Since this scenario requires overlapping on both sides, $x$ can range from 2 to $u = \min(C, 2\lambda - 2)$. The summation has the form:

$$\sum_{x=2}^{u}(-|x-\lambda| + \lambda - 1)(k-1)\binom{w+x-1}{k-1}$$

$$= \sum_{x=2}^{\lambda}(x-1)(k-1)\binom{w+x-1}{k-1} + \sum_{x=\lambda+1}^{u}(2\lambda - x - 1)(k-1)\binom{w+x-1}{k-1}$$

$$= (k-1)\sum_{x=w+1}^{w+\lambda-1}(x-w)\binom{x}{k-1} + (k-1)\sum_{x=w+\lambda}^{w+u-1}(2\lambda - x + w - 2)\binom{x}{k-1}$$

$$= (k-1)\sum_{x=w+1}^{w+\lambda-1}x\binom{x}{k-1} - (k-1)w\sum_{x=w+1}^{w+\lambda-1}\binom{x}{k-1} - (k-1)\sum_{x=w+\lambda}^{w+u-1}x\binom{x}{k-1}$$

$$+ (k-1)(2\lambda + w - 2)\sum_{x=w+\lambda}^{w+u-1}\binom{x}{k-1}$$

$$= (k-1)\sum_{x=w+1}^{w+\lambda-1}\left[(k-1)\binom{x}{k-1} + k\binom{x}{k}\right] - (k-1)w\sum_{x=w+1}^{w+\lambda-1}\binom{x}{k-1}$$

$$- (k-1)\sum_{x=w+\lambda}^{w+u-1}\left[(k-1)\binom{x}{k-1} + k\binom{x}{k}\right] + (k-1)(2\lambda + w - 2)\sum_{w+\lambda}^{w+u-1}\binom{x}{k-1}$$

$$= (k-1)(k-1-w)\sum_{x=w+1}^{w+\lambda-1}\binom{x}{k-1} + k(k-1)\sum_{x=w+1}^{w+\lambda-1}\binom{x}{k}$$

$$+ (k-1)(2\lambda + w - 2 - k + 1)\sum_{x=w+\lambda}^{w+u-1}\binom{x}{k-1} - k(k-1)\sum_{x=w+\lambda}^{w+u-1}\binom{x}{k}$$

$$= (k-1)(k-1-w)\left[\binom{w+\lambda}{k} - \binom{w+1}{k}\right] + k(k-1)\left[\binom{w+\lambda}{k+1} - \binom{w+1}{k+1}\right]$$

$$+ (k-1)(2\lambda + w - 2 - k + 1)\left[\binom{w+u}{k} - \binom{w+\lambda}{k}\right] - k(k-1)\left[\binom{w+u}{k+1} - \binom{w+\lambda}{k+1}\right]$$

$$= \frac{k-1}{k+1}(2k\lambda + 2\lambda + w - 2k - ku - 1)\binom{w+u}{k} - 2(k-1)\binom{w+\lambda}{k+1} + (k-1)\binom{w+1}{k+1}$$

When we combine Scenarios 1 to 3, we have

$$(k+1)\binom{w+1}{k+1} + 2k\binom{w+\lambda}{k+1} - 2k\binom{w+1}{k+1}$$

$$+ \frac{k-1}{k+1}(2k\lambda + 2\lambda + w - 2k - ku - 1)\binom{w+u}{k} - 2(k-1)\binom{w+\lambda}{k+1} + (k-1)\binom{w+1}{k+1}$$

$$= 2\binom{w+\lambda}{k+1} + \frac{k-1}{k+1}(2k\lambda + 2\lambda + w - 2k - ku - 1)\binom{w+u}{k}$$

The total number of possible combinations is given by

$$\binom{L - k\lambda + k}{k}(L - C + 1)$$

The cover probability is given by

$$1 - \frac{2\binom{w+\lambda}{k+1} + \frac{k-1}{k+1}(2k\lambda + 2\lambda + w - 2k - ku - 1)\binom{w+u}{k}}{\binom{L-k\lambda+k}{k}(L - C + 1)}$$

### A.2 EXAMPLE PLOT

We consider a hypothetical problem whose entire context length $L = 50$, and we use an observation span length $C = 5$. We show $\pi(\lambda, k; L = 50, C = 5)$, i.e. the probability that the oracle model correctly answers the problem (i.e. a "1" outcome), on the $\lambda$-k plane in Figure 3(a), as well as the probability that the oracle model cannot answer the problem (i.e. an "IDK" outcome) in Figure 3(b).

(a) Probability observing "1" ($\pi(\lambda, k; L = 50, C = 5)$)

(b) Probability observing "IDK" ($1 - \pi(\lambda, k; L = 50, C = 5)$)

Figure 3: Probability that the oracle model correctly answers the problem (i.e. a "1" outcome) and cannot answer the problem (i.e. an "IDK" outcome) under the COW assumption.

## B  DETAILS OF $\rho$ IN THE PIG SCENARIO ($k$-REPEATED $\lambda$ LENGTH-1 ASPECTS)

### B.1 FORMULA & DERIVATION

The cover probability for the PIG assumption ($k$-repeated $\lambda$ length-1 aspects) $\tilde{\rho}(s, \lambda, k; L, C)$ is given by:

$$\tilde{\rho}(s, \lambda, k; L, C) = \frac{\binom{\lambda}{s\lambda}}{\binom{L}{C}} \sum_{t=0}^{\lfloor \min(s\lambda, d) \rfloor} (-1)^t m_t$$

where

$$d = \frac{L - C}{k} - (1 - s)\lambda$$

$$m_t = \binom{(d - t)k + C}{C}\binom{s\lambda}{t}$$

We now give one derivation using the inclusion-exclusion principle.

**Derivation.** First, we put uncovered $k(1 - s)\lambda$ segments into the sequence outside of the observation, which has a length of $L - C$. Each aspect has $k$ repeats, and thus $k!$ duplicate counts. The total unique count is given by

$$\binom{L - C}{k(1 - s)\lambda} \frac{(k(1 - s)\lambda)!}{(k!)^{(1-s)\lambda}}$$

Then, we put $s\lambda$ covered aspects onto the entire context of length $L$, excluding the occupied $k(1-s)\lambda$ positions. The total unique count is given by

$$\binom{L-k(1-s)\lambda}{ks\lambda}\frac{(ks\lambda)!}{(k!)^{s\lambda}}$$

There exist invalid allocations. In fact, we need to make sure each one of $s\lambda$ aspects should appear in the observation span at least once. We can count the number of combinations that a given aspect only occurs outside of the observation. Since there are $k$ occurrences, it is given by

$$\binom{L-C-k(1-s)\lambda}{k}\binom{L-k(1-s)\lambda-k}{k(s\lambda-1)}\frac{(k(s\lambda-1))!}{(k!)^{s\lambda-1}}$$

We can alternate the aspect from one of $s\lambda$ covered aspects, so the total number of invalid combinations is

$$\binom{L-C-k(1-s)\lambda}{k}\binom{L-k(1-s)\lambda-k}{k(s\lambda-1)}\frac{(k(s\lambda-1))!}{(k!)^{s\lambda-1}}\binom{s\lambda}{1}$$

It "over-counts" when two aspects both occur outside of the observation, whose total number is given by

$$\binom{L-C-k(1-s)\lambda}{2k}\frac{(2k)!}{(k!)^2}\binom{L-k(1-s)\lambda-2k}{k(s\lambda-2)}\frac{(k(s\lambda-2))!}{(k!)^{s\lambda-2}}\binom{s\lambda}{2}$$

Using the inclusion-exclusion formula, we can derive the actual total count, which is given by

$$\sum_{i=0}^{\lfloor\min(s\lambda,d)\rfloor}(-1)^i\binom{L-C-k(1-s)\lambda}{ik}\frac{(ik)!}{(k!)^i}\binom{L-k(1-s)\lambda-ik}{k(s\lambda-i)}\frac{(k(s\lambda-i))!}{(k!)^{s\lambda-i}}\binom{s\lambda}{i}$$

$$=\frac{(L-C-k(1-s)\lambda)!C!}{(k!)^{s\lambda}(L-k\lambda)!}\sum_{i=0}^{\lfloor\min(s\lambda,d)\rfloor}(-1)^i\binom{L-k(1-s)\lambda-ik}{C}\binom{s\lambda}{i}$$

where $d=\frac{L-C}{k}-(1-s)\lambda$.

Then, there are $\binom{\lambda}{s\lambda}$ ways to choose which aspects are covered. Finally, the total number of possible combinations is given by

$$\binom{\lambda}{k\lambda}\frac{(k\lambda)!}{(k!)^{\lambda}}$$

We put them all together to obtain the cover probability of $\tilde{\rho}$, which is given by

$$\binom{L-C}{k(1-s)\lambda}\frac{(k(1-s)\lambda)!}{(k!)^{(1-s)\lambda}}\frac{(L-C-k(1-s)\lambda)!C!}{(k!)^{s\lambda}(L-k\lambda)!}\sum_{i=0}^{\lfloor\min(s\lambda,d)\rfloor}(-1)^i\binom{L-k(1-s)\lambda-ik}{C}\binom{s\lambda}{i}$$

$$\binom{\lambda}{s\lambda}\left[\binom{\lambda}{k\lambda}\frac{(k\lambda)!}{(k!)^{\lambda}}\right]^{-1}$$

$$=\frac{\binom{\lambda}{s\lambda}}{\binom{L}{C}}\sum_{i=0}^{\lfloor\min(s\lambda,d)\rfloor}\binom{L-k(1-s)\lambda-ik}{C}\binom{s\lambda}{i}$$

## B.2 Linear Interpolation of $\rho(s,\lambda,k;L,C)$

The function $\tilde{\rho}$ only takes discrete values, i.e. $s$ is a multiple of $1/\lambda$. To further incorporate any arbitrary proportion $s$, we need to further define $s$ between $i/\lambda$ and $(i+1)/\lambda$ for any $i$. We use linear interpolation and provide the formula for $\rho(s_{ij}^{\mathcal{O}},\lambda_i,k_i;L_i,C_{ij})$ as follows:

$$\rho(s,\lambda,k;L,C)=\begin{cases}(\lambda+1)\tilde{\rho}(s,\lambda,k;L,C) & \text{if } s\in\frac{\mathbb{N}}{\lambda}\\(\lambda^2+\lambda)\left[(b-s)\tilde{\rho}(a,\lambda,k;L,C)+(s-a)\tilde{\rho}(b,\lambda,k;L,C)\right] & \text{o.w.}\end{cases}$$

where

$$a = \frac{\lfloor s\lambda \rfloor}{\lambda} \quad \text{and} \quad b = \frac{\lceil s\lambda \rceil}{\lambda}$$

In order to make it a probability density function, we also need to integrate $\rho$ over $s$, which we can hardly find a closed form solution for. Instead, we simply multiply $\pi$ by $\lambda + 1$ to approximate. This approximation can be inaccurate when $\lambda$ is small. However, when we estimate $\lambda$ and $k$, we will compare the probability density function (pdf) of the oracle model to that of the noisy model. We make a uniform prior assumption, and the probability mass function (pmf) of the discrete uniform distribution and the probability density function of the continuous variant differ by a factor of $\lambda + 1$.

### B.3 EXAMPLE PLOT

We consider the same context as in Appendix A.2, with the entire context length $L = 50$ and the observation length $C = 5$. We show $\rho(\lambda, k; s = 1, L = 50, C = 5)$, $\rho(\lambda, k; s = 0.5, L = 50, C = 5)$, and $\rho(\lambda, k; s = 0, L = 50, C = 5)$, i.e. the probability that the oracle model observes a partial point of 1, 0.5, or 0 respectively, on the $\lambda$-k plane in Figure 4.

We first see that $\rho(\lambda, k; s = 1, L = 50, C = 5)$ has a very similar probability distribution as $\pi(\lambda, k; L = 50, C = 5)$ in Figure 3. The probability peaks when $\lambda = 2$ and $k = 20$ in both cases. Then, when we compare between the probability of observing "IDK" under the COW assumption $(1 - \pi(\lambda, k; L = 50, C = 5))$ and the probability of observing "0" under the PIG assumption $(\rho(\lambda, k; s = 0, L = 50, C = 5))$, although they also appear similar, there is some subtle difference. The most notable difference lies around the area of large $\lambda$. As $\lambda$ increases, the probability of observing "IDK" under the COW assumption also increases monotonically (before it falls into the invalid area $k\lambda > L$), but the probability of observing "0" under the PIG assumption first increases and then decreases to zero. In fact, with the "IDK" / "0" outcome, the COW assumption believes the problem is very hard and requires a ground-truth context ($\lambda$) longer than the current observation length to answer it. The PIG assumption, on the other hand, believes there are not that many aspects ($\lambda$) in the context. Otherwise, the sample should at least get some partial point, not zero.

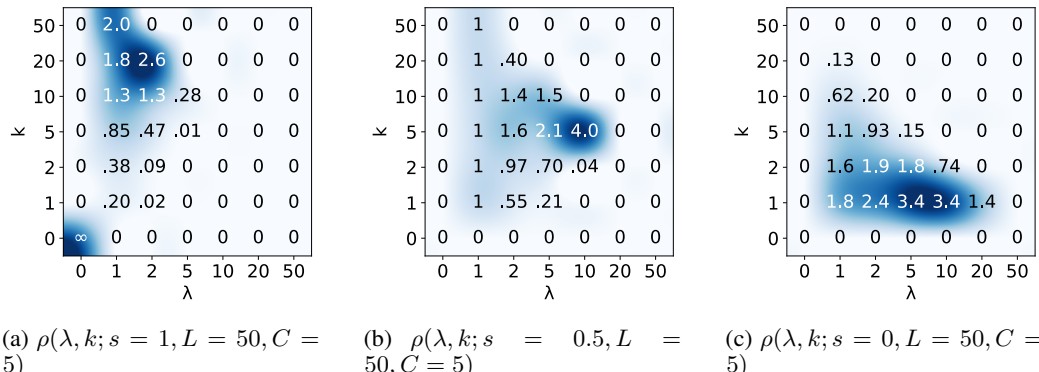

(a) $\rho(\lambda, k; s = 1, L = 50, C = 5)$

(b) $\rho(\lambda, k; s = 0.5, L = 50, C = 5)$

(c) $\rho(\lambda, k; s = 0, L = 50, C = 5)$

Figure 4: Probability that the oracle model observes a proportion at 1, 0.5 and 0 respectively, under the PIG assumption.

## C PSEUDOCODE OF EM-BASED MLE ALGORITHM

### C.1 COW SCENARIO

We first list the EM-based MLE algorithm for DOLCE in the COW Scenario in Algorithm 1.

### C.2 PIG SCENARIO

We then list the EM-based MLE algorithm for DOLCE in the PIG Scenario in Algorithm 2.

---

**Algorithm 1** EM-based MLE algorithm for DOLCE in the COW Scenario

---

**Require:** Number of steps: $T > 0$
**Require:** Candidate $\lambda$ set: $\Lambda$
**Require:** Candidate $k$ set: $K$
**Require:** Problem index set: $I$
**Require:** Full context length for problem $i \in I$: $L_i$
**Require:** Observed evaluation outcome index set for $i \in I$: $J_i$
**Require:** Observation span length for observation $i \in I, j \in J$: $C_{ij}$
**Require:** Observed evaluation outcome for $i \in I, j \in J$: $x_{ij} \in \{1, 0, \emptyset\}$
**Ensure:** Optimal $\lambda_i^*$ and $k_i^*$ for each $i \in I$
  **for all** $i \in I, j \in J_i, x \in \{1, 0, \emptyset\}, z \in \{\mathcal{O}, \mathcal{N}\}$ **do**          $\triangleright$ Initialize $q(\mathrm{z}|\mathrm{x})$.
    $q_{i,j}(\mathrm{z} = z | \mathrm{x} = x) \leftarrow 0.5$
  **end for**
  **for** $t = 1, \ldots, T$ **do**                                                          $\triangleright$ Main loop.
    **for all** $x \in \{1, 0, \emptyset\}$ **do**                              $\triangleright$ Update $P(\mathrm{x} = x | \mathrm{z} = \mathcal{N})$ in M-step.

$$P(\mathrm{x} = x | \mathrm{z} = \mathcal{N}) \leftarrow \frac{\sum_{i,j} q_{i,j}(\mathrm{z} = \mathcal{N} | \mathrm{x} = x) [\![x_{ij} = x]\!]}{\sum_{i,j,x'} q_{i,j}(\mathrm{z} = \mathcal{N} | \mathrm{x} = x') [\![x_{ij} = x']\!]}$$

    **end for**
    **for all** $i \in I$ **do**                                          $\triangleright$ Update $P(\mathrm{x} | \mathrm{z} = \mathcal{O})$ in M-step.
      **for all** $\lambda \in \Lambda$ **do**
        **for all** $x \in \{1, 0, \emptyset\}$ **do**

$$P_{\mathrm{nonpar},i}(\mathrm{x} = x | \mathrm{z} = \mathcal{O}; \lambda) \leftarrow \frac{\sum_j q_{i,j}(\mathrm{z} = \mathcal{O} | \mathrm{x} = x) [\![x_{ij} = x, C_{ij} < \lambda]\!]}{\sum_{j,x'} q_{i,j}(\mathrm{z} = \mathcal{O} | \mathrm{x} = x') [\![x_{ij} = x', C_{ij} < \lambda]\!]}$$

        **end for**
        **for all** $k \in K, j \in J_i$ **do**
          $P_{\mathrm{par},i,j}(\mathrm{x} = 1 | \mathrm{z} = \mathcal{O}; \lambda, k) \leftarrow \pi(\lambda, k; L_i, C_{ij})^{[\![x_{ij} = 1]\!]}$          $\triangleright$ Eq. 3
          $P_{\mathrm{par},i,j}(\mathrm{x} = 0 | \mathrm{z} = \mathcal{O}; \lambda, k) \leftarrow 0^{[\![x_{ij} = 0]\!]}$
          $P_{\mathrm{par},i,j}(\mathrm{x} = \emptyset | \mathrm{z} = \mathcal{O}; \lambda, k) \leftarrow (1 - \pi(\lambda, k; L_i, C_{ij}))^{[\![x_{ij} = \emptyset]\!]}$
          **for all** $x \in \{1, 0, \emptyset\}$ **do**
            $P_{i,j}(\mathrm{x} = x | \mathrm{z} = \mathcal{O}; \lambda, k) \leftarrow \big[ P_{\mathrm{nonpar},i}(\mathrm{x} = x | \mathrm{z} = \mathcal{O}; \lambda)^{[\![x_{ij} = x, C_{ij} < \lambda]\!]}$     $\triangleright$ Eq. 4
            $\qquad P_{\mathrm{par},i,j}(\mathrm{x} = x | \mathrm{z} = \mathcal{O}; \lambda, k)^{[\![x_{ij} = x, C_{ij} \geq \lambda]\!]} \big]$
          **end for**
        **end for**
      **end for**
      $\lambda_i^*, k_i^* \leftarrow \arg\max_{\lambda, k} \prod_j [P_{i,j}(\mathrm{x} = x_{ij} | \mathrm{z} = \mathcal{O}; \lambda, k) P_i(\mathrm{z} = \mathcal{O})$     $\triangleright$ Brute-force search.
      $\qquad + P(\mathrm{x} = x_{ij} | \mathrm{z} = \mathcal{N}) P_i(\mathrm{z} = \mathcal{N})]$
      **for all** $x \in \{1, 0, \emptyset\}, j \in J_i$ **do**
        $P_{i,j}(\mathrm{x} = x | \mathrm{z} = \mathcal{O}) \leftarrow P_{i,j}(\mathrm{x} = x | \mathrm{z} = \mathcal{O}; \lambda_i^*, k_i^*)$
      **end for**
    **end for**
    **for all** $i \in I, z \in \{\mathcal{N}, \mathcal{O}\}$ **do**                              $\triangleright$ Update $P(\mathrm{z})$ in M-step.

$$P_i(\mathrm{z} = z) \leftarrow \frac{\sum_{j,x'} q_{i,j}(\mathrm{z} = z | \mathrm{x} = x') [\![x_{ij} = x]\!]}{|J_i|}$$

    **end for**
    **for all** $i \in I, j \in J_i, x \in \{1, 0, \emptyset\}$ **do**                    $\triangleright$ Update $q(\mathrm{z}|\mathrm{x})$ in E-step.

$$q_{i,j}(\mathrm{z} = \mathcal{N} | \mathrm{x} = x) \leftarrow \frac{P(\mathrm{x} = x | \mathrm{z} = \mathcal{N}) P_i(\mathrm{z} = \mathcal{N})}{P(\mathrm{x} = x | \mathrm{z} = \mathcal{N}) P_i(\mathrm{z} = \mathcal{N}) + P_{i,j}(\mathrm{x} = x | \mathrm{z} = \mathcal{O}) P_i(\mathrm{z} = \mathcal{O})}$$

$$q_{i,j}(\mathrm{z} = \mathcal{O} | \mathrm{x} = x) \leftarrow \frac{P_{i,j}(\mathrm{x} = x | \mathrm{z} = \mathcal{O}) P_i(\mathrm{z} = \mathcal{O})}{P(\mathrm{x} = x | \mathrm{z} = \mathcal{N}) P_i(\mathrm{z} = \mathcal{N}) + P_{i,j}(\mathrm{x} = x | \mathrm{z} = \mathcal{O}) P_i(\mathrm{z} = \mathcal{O})}$$

    **end for**
  **end for**

---

---

**Algorithm 2** EM-based MLE algorithm for DOLCE in the PIG Scenario

---

**Require:** Number of steps: $T > 0$
**Require:** Candidate $\lambda$ set: $\Lambda$
**Require:** Candidate $k$ set: $K$
**Require:** Problem index set: $I$
**Require:** Full context length for problem $i \in I$: $L_i$
**Require:** Observed evaluation outcome index set for $i \in I$: $J_i$
**Require:** Observation span length for observation $i \in I, j \in J$: $C_{ij}$
**Require:** Observed evaluation outcome for $i \in I, j \in J$: $s_{ij} \in [0,1] \cup \{\emptyset\}$
**Ensure:** Optimal $\lambda_i^*$ and $k_i^*$ for each $i \in I$
  **for all** $i \in I, j \in J_i$ **do**                                ▷ Initialize discrete y from continuous $s$.
    **if** $s_{ij} = \emptyset$ **then** $y_{ij}(1) \leftarrow 0, y_{ij}(\emptyset) \leftarrow 1, y_{ij}(0) \leftarrow 0$
    **else** $y_{ij}(1) \leftarrow s_{ij}, y_{ij}(\emptyset) \leftarrow 0, y_{ij}(0) \leftarrow 1 - s_{ij}$
    **end if**
  **end for**
  **for all** $i \in I, j \in J_i, y \in \{1, 0, \emptyset\}, z \in \{\mathcal{O}, \mathcal{N}\}$ **do**               ▷ Initialize $q(\mathrm{z}|\mathrm{y})$.
    $q_{i,j}(\mathrm{z} = z | \mathrm{y} = y) \leftarrow 0.5$
  **end for**
  **for** $t = 1, \ldots, T$ **do**                                              ▷ Main loop.
    **for all** $y \in \{1, 0, \emptyset\}$ **do**                   ▷ Update $P(\mathrm{y} = y | \mathrm{z} = \mathcal{N})$ in M-step.

$$P(\mathrm{y} = y | \mathrm{z} = \mathcal{N}) \leftarrow \frac{\sum_{i,j} q_{i,j}(\mathrm{z} = \mathcal{N} | \mathrm{y} = y) y_{ij}(y)}{\sum_{i,j,y'} q_{i,j}(\mathrm{z} = \mathcal{N} | \mathrm{y} = y') y_{ij}(y')}$$

    **end for**
    **for all** $i \in I$ **do**                            ▷ Update $P(\mathrm{y} | \mathrm{z} = \mathcal{O})$ in M-step.
      **for all** $\lambda \in \Lambda$ **do**
        **for all** $y \in \{1, 0, \emptyset\}, c \in$ unique $\{C_{ij}\}_j$ **do**

$$P_{i,c}(\mathrm{y} = y | \mathrm{z} = \mathcal{O}) \leftarrow \frac{\sum_j q_{i,j}(\mathrm{z} = \mathcal{O} | \mathrm{y} = y) y_{ij}(y) [\![C_{ij} = c]\!]}{\sum_{j,y'} q_{i,j}(\mathrm{z} = \mathcal{O} | \mathrm{y} = y') y_{ij}(y') [\![C_{ij} = c]\!]}$$

        **end for**
        **for all** $k \in K, j \in J_i$ **do**
          $p_{i,j}(\mathrm{x} = s_{ij} | \mathrm{z} = \mathcal{O}; \lambda, k) \leftarrow \rho(P_{i,C_{ij}}(\mathrm{y} = 1 | \mathrm{z} = \mathcal{O}), \lambda, k; L_i, C_{ij})$     ▷ Eq. 6
          $p_{i,j}(\mathrm{x} = s_{ij} | \mathrm{z} = \mathcal{N}; \lambda, k) \leftarrow 1$
        **end for**
      **end for**
      $\lambda_i^*, k_i^* \leftarrow \arg\max_{\lambda,k} \prod_j \sum_z [p_{i,j}(\mathrm{x} = s_{ij} | \mathrm{z} = z; \lambda, k) P_i(\mathrm{z} = z)]$   ▷ Brute-force search.
      **for all** $c \in$ unique $\{C_{ij}\}_j$ **do**
        $P_{i,c}(\mathrm{y} = 1 | \mathrm{z} = \mathcal{O}) \leftarrow \arg\max_s \rho(s, \lambda_i^*, k_i^*; L_i, c)$
        **for all** $y \in \{0, \emptyset\}$ **do**

$$P_{i,c}(\mathrm{y} = y | \mathrm{z} = \mathcal{O}) \leftarrow \frac{[1 - P_{i,c}(\mathrm{y} = 1 | \mathrm{z} = \mathcal{O})] P_{i,c}(\mathrm{y} = y | \mathrm{z} = \mathcal{O})}{\sum_{y' \in \{0, \emptyset\}} P_{i,c}(\mathrm{y} = y' | \mathrm{z} = \mathcal{O})}$$

        **end for**
      **end for**
    **end for**
    **for all** $i \in I, z \in \{\mathcal{N}, \mathcal{O}\}$ **do**                       ▷ Update $P(\mathrm{z})$ in M-step.

$$P_i(\mathrm{z} = z) \leftarrow \frac{\sum_{j,y'} q_{i,j}(\mathrm{z} = z | \mathrm{y} = y') y_{ij}(y)}{|J_i|}$$

    **end for**
    **for all** $i \in I, j \in J_i, y \in \{1, 0, \emptyset\}$ **do**               ▷ Update $q(\mathrm{z}|\mathrm{y})$ in E-step.

$$q_{i,j}(\mathrm{z} = \mathcal{N} | \mathrm{y} = y) \leftarrow \frac{P(\mathrm{y} = y | \mathrm{z} = \mathcal{N}) P_i(\mathrm{z} = \mathcal{N})}{P(\mathrm{y} = y | \mathrm{z} = \mathcal{N}) P_i(\mathrm{z} = \mathcal{N}) + P_{i,C_{ij}}(\mathrm{y} = y | \mathrm{z} = \mathcal{O}) P_i(\mathrm{z} = \mathcal{O})}$$

$$q_{i,j}(\mathrm{z} = \mathcal{O} | \mathrm{y} = y) \leftarrow \frac{P_{i,C_{ij}}(\mathrm{y} = y | \mathrm{z} = \mathcal{O}) P_i(\mathrm{z} = \mathcal{O})}{P(\mathrm{y} = y | \mathrm{z} = \mathcal{N}) P_i(\mathrm{z} = \mathcal{N}) + P_{i,C_{ij}}(\mathrm{y} = y | \mathrm{z} = \mathcal{O}) P_i(\mathrm{z} = \mathcal{O})}$$

    **end for**
  **end for**

---

## D  SUITE & TASK SPECIFIC PREPROCESSING

We describe the task specific preprocessing steps in each benchmark suite in this section.

**L-Eval.** We use the "input" as the context, then flatten the "instructions" to create multiple independent problems.

**BAMBOO.** We exclude the two sorting tasks, since they require special assumptions and sampling based methods. Some of the BAMBOO tasks have a single "content" field consisting of instructions, contexts, and questions. We need to identify individual parts from the "content", since we can only sample from the context while keep the entire instruction and question available to the model in all samples. In particular,

- For an **AltQA** problem, we split the "content" into a question from the "Question" section, a context from the "Document" section, and an actual instruction from the remaining content.
- For **ShowsPred** and **MeetingPred** tasks, we treat the last sentence as the question, and the prior conversation as the context.
- For **PrivateEval** task, we use the lines between "# [start]" and "# [end]" as the context, and the lines after "# [end]" and the question.

**LongBench.** We use the English subsets of this multilingual dataset. We use the original "input" and "context" fields.

## E  PROMPTS & IDK INSTRUCTION

For all tasks, we reuse the instructions provided with the official distributions of benchmark suites. Then, we extend the instructions in all but summarization tasks to allow the model to generate "IDK" ("unanswerable" or "E" for a four-choice question) whenever needed, similar to the pre-existing instruction for the LongBench Qasper task. Examples include:

```
If you cannot answer the question, you should answer "Unanswerable".
```

and

```
If the question cannot be answered based on the information in the article, write
"E" for "unanswerable"
...
E. The question is unanswerable.
```

## F  EXAMPLE OF LENGTH STATISTICS & UNIT IDENTIFICATION

We are not restricted from using sentences as the units. In fact, we may choose different unit granularities, e.g. tokens or even characters at one extreme and one or several full document(s) at the other extreme. The problem with the former setup is that we may end up with a large number of spans that are not semantically coherent, which wastes our computation resource. The problem with the latter is that we may still present irrelevant contexts to the model and get inflated numbers, which also challenges the long context capability of the test model. A general rule of thumb is to choose a unit granularity such that the derived units with which the context lengths have little variance when measured in number of characters or tokens.

We have developed tools to help analyze the distribution of lengths and numbers of spans using different unit granularities. An example for L-Eval NQ task is shown in Figure 5.

## G  UNIT GRANULARITIES & PREPROCESSING STEPS

For most tasks, we choose to use sentences, and in other cases (e.g. legal, scientific, or coding tasks), we also use paragraphs. In some special cases, e.g. in-context few-shot problems, we

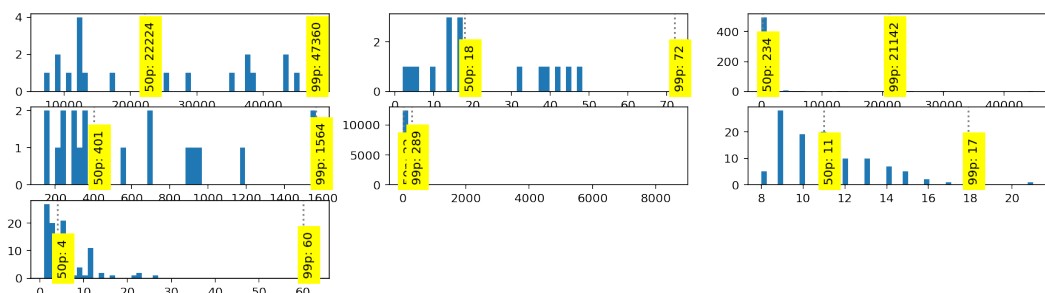

Figure 5: Example of a length statistics for L-Eval NQ task, where it shows different length distributions across all problems: (1) number of tokens for each input, (2) number of units if the contexts are split by the "<P>" tags, (3) number of tokens in each unit if the contexts are split by the "<P>" tag, (4) number of units if the inputs are split into sentences as identified by NLTK, (5) number of tokens in each unit if the inputs are split into sentences as identified by NLTK, (6) number of tokens in each instruction, and (7) number of tokens in each ground-truth answer. 50-th and 99-th percentiles are also annotated.

consider each shot, consisting of an input and an expected output, is an atomic unit of the context. We summarize our decisions of unit granularities in this section. We define a set of unit patterns, including single linebreaks, multi-linebreaks, NLTK-identified sentences, "<P>" and "</P>" pairs, as well as special identifiers, e.g. "[scene NUMBER]", "Review #", "Passage NUMBER", "Passage:"/"Question:"/"Answer:" triplets, "Dialogue:"/"Summary:" pairs, etc, as well as some preprocessing tools, including converting commas to periods (as TTS transcripts use only commas). We list the specs selected in the experiment with their detailed descriptions in Table 3. For example, we drop the explicit "[scene NUMBER]" marker, since the length of the scenes varies largely.

We list the task-specific preprocessing steps as well as other configuration specifications in Table 4. We select the observation lengths (i.e. number of units) exponentially. We stop increasing the observation length when the maximum context length across all problems is below the observation length, or the sampled span lengths start to exceed 16K, which we do not consider "short" any more. We also need to take the instruction, question, as well as the maximum output length into the computation of the maximum length.

We also show the answer extractor. We aim to find the answer span that matches the expected ground truth answer format, due to lack of a human or oracle evaluator. For most ROUGE-evaluated tasks, we take the entire output as the predicted answer. For accuracy and F-1 evaluated tasks, we extract the answer phrase and skip other reasoning or commenting parts of the output. We note that if the prompt has a clear format instruction, we impose little postprocessing. We notice this can happen in some tasks, which we will discuss in Appendix K.3.

Table 3: Spec details for the unit preprocessing selected in the experiments.

| SPEC | DETAIL | SPLIT-BY REGEX |
|------|--------|----------------|
| b | Blocks | Multi-line breaks `(?:\n *){2,}` |
| c | Every two lines | |
| l | Lines | `\n` |
| n | Reviews | `Review #\d+` |
| o | Passage | `Passage \d+:` |
| q | Passage/question/answer triplets | `Passage:.*?Question:.*?Answer:\n.*?\n` |
| r | Replace commas with periods. | |
| s | NLTK identified sentences | |
| t | NLTK identified sentences from pretokenized inputs | |
| u | Dialogue/summary pairs | `Dialogue:.*?Summary:.*?\n` |

Table 4: Preprocessing and postprocessing specs for the tasks.

| SUITE | TASK | UNIT | OBSERVATION LENGTHS | ANSWER EXTRACTOR | METRIC |
|---|---|---|---|---|---|
| L-Eval | TOEFL (Tseng et al., 2016) | rlt | 0, 1, 2, 5, 10, 20, 50, 100, $L$ | Extract first word | Accuracy |
| | GSM (Cobbe et al., 2021; An et al., 2024) | b | 0, 1, 2, 5, 10, $L$ | Extract numeric answer | Accuracy |
| | QuALITY (Pang et al., 2022; An et al., 2024) | s | 0, 1, 2, 5, 10, 20, 50, 100, $L$ | Extract 4-choice answer | Accuracy |
| | Coursera (An et al., 2024) | ls | 0, 1, 2, 5, 10, 20, 50, 100, 200, $L$ | Extract 4-choice answer | Accuracy |
| | TopicRet (Li et al., 2023; An et al., 2024) | l | 1, 2, 5, 10, 20, 50, 100, $L$ | Take first line | Accuracy |
| | SFcition (An et al., 2024) | s | 1, 2, 5, 10, 20, 50, 100, 200, 500, $L$ | Take first line | Accuracy |
| | CodeU (An et al., 2024) | l | 0, 1, 10, 20, 50, 100, 200, 500 | Extract coding answer | Accuracy |
| | | b | 0, 1, 2, 5, 10, 20, 50 | | |
| | MultiDoc2Dial (Feng et al., 2021) | b | 0, 1, 2, 5, 10, 20, $L$ | None | F1 |
| | Qasper (Dasigi et al., 2021) | ls | 0, 1, 2, 5, 10, 20, 50, $L$ | None | F1 |
| | LongFQA (An et al., 2024) | ls | 0, 1, 2, 5, 10, 20, 50, 100, $L$ | None | F1 |
| | NQ (Kwiatkowski et al., 2019) | t | 0, 1, 2, 5, 10, 20, 50, 100 | None | F1 |
| | CUAD (Hendrycks et al.) | b | 0, 1, 2, 5, 10, 20 | None | F1 |
| | NarrativeQA (Kočiský et al., 2018) | s | 0, 1, 2, 5, 10, 20, 50, 100, 200 | None | F1 |
| | MultiNews (Fabbri et al., 2019) | s | 0, 1, 2, 5, 10, 20, 50, $L$ | None | RougeL |
| | GovReport (Huang et al., 2021) | ls | 0, 1, 2, 5, 10, 20, 50 | None | RougeL |
| | BigPatent (Sharma et al., 2019) | ls | 0, 1, 2, 5, 10, 20, 50, 100, $L$ | None | RougeL |
| | SummScreen (Chen et al., 2022) | ls | 0, 1, 2, 5, 10, 20, 50, 100, 200, 500, $L$ | None | RougeL |
| | OpenReview (An et al., 2024) | ls | 0, 1, 2, 5, 10, 20, 50, 100, $L$ | None | RougeL |
| | QMSum (Zhong et al., 2021) | l | 0, 1, 2, 5, 10, 20, 50 | None | RougeL |
| | SPACE (Angelidis et al., 2021; An et al., 2024) | n | 0, 1, 2, 5, 10, 20, $L$ | None | RougeL |
| BAMBOO | AltQA (Dong et al., 2024) | lt | 0, 1, 2, 5, 10, 20, 50, 100 (4K only), $L$ | Extract answer | Accuracy |
| | PaperQA (Dong et al., 2024) | ls | 0, 1, 2, 5, 10, 20, 50, 100, $L$ | Extract 4-choice answer | Accuracy |

| | Dataset | | Values | Extract | Metric |
|---|---|---|---|---|---|
| | MeetingQA (Dong et al., 2024) | b | 0, 1, 2, 5, 10, 20, 50 (16K only), 100 (16K only), $L$ | Extract 4-choice answer | Accuracy |
| | SenHallu (Dong et al., 2024) | ls | 0, 1, 2, 5, 10, 20, 50, 100, 200 (16K only), $L$ | Extract answer | F1 |
| | AbsHallu (Dong et al., 2024) | ls | 0, 1, 2, 5, 10, 20, 50, 100, 200 (16K only), $L$ | Extract answer | F1 |
| | ShowsPred (Dong et al., 2024) | l | 1, 2, 5, 10, 20, 50, 100 (16K only), $L$ | Take first line | Accuracy |
| | MeetingPred (Dong et al., 2024) | l | 1, 2, 5, 10, 20, 50, 100 (16K only), $L$ | Take first line | Accuracy |
| | PrivateEval (Dong et al., 2024) | l | 1, 2, 5, 10, 20, 50, 100 (16K only), $L$ | None | RougeL |
| | | b | 1, $L$ | | |
| LongBench | NarrativeQA (Kočiský et al., 2018) | b | 0, 1, 2, 5, 10, 20, 50 | Extract answer | F1 |
| | Qasper (Dasigi et al., 2021) | ls | 0, 1, 2, 5, 10, 20, 50, 100, 200, $L$ | None | F1 |
| | MultiFieldQA (Bai et al., 2024) | ls | 0, 1, 2, 5, 10, 20, 50, 100, $L$ | None | F1 |
| | HotpotQA (Yang et al., 2018) | b | 0, 1, 2, 5, 10, $L$ | Extract answer | F1 |
| | 2WikiMultihopQA (Ho et al., 2020) | b | 0, 1, 2, 5, 10, $L$ | Extract answer | F1 |
| | | o | 0, 1, 2, 5, $L$ | | |
| | MuSiQue (Trivedi et al., 2022) | b | 0, 1, 2, 5, 10, $L$ | Extract answer | F1 |
| | | o | 0, 1, 2, 5, 10, $L$ | | |
| | GovReport (Huang et al., 2021) | s | 1, 10, 20, 50, 100 | None | RougeL |
| | QMSum (Zhong et al., 2021) | lt | 1, 10, 20, 50, 100, 200 | None | RougeL |
| | MultiNews (Fabbri et al., 2019) | ls | 1, 10, 20, 50, 100, $L$ | None | RougeL |
| | | o | 1, 2, 5, $L$ | | |
| | TREC Li & Roth (2002) | c | 0, 1, 2, 5, 10, 20, 50, 100, 200, $L$ | Extract TREC answer | Accuracy |
| | TriviaQA (Joshi et al., 2017) | q | 0, 1, 2, 5, 10, $L$ | Extract answer | F1 |
| | SAMSum (Gliwa et al., 2019) | u | 0, 1, 2, 5, 10, 20, $L$ | None | RougeL |
| | PassageCount (Bai et al., 2024) | b | 1, 2, 5, 10, 20, $L$ | None | Accuracy |
| | PassageRetrieval (Bai et al., 2024) | b | 1, 2, 5, 10, 20, $L$ | Take first line | Accuracy |
| | LCC (Guo et al., 2023) | l | 1, 10, 20, 50, 100, 200, $L$ | None | EditSim |
| | RepoBench-p (Liu et al., 2024d) | b | 0, 1, 10, 20 | Extract answer | EditSim |
| | | l | 0, 1, 100, 200 | | |

## H  HARTIGANS' DIP TEST RESULTS

In this section, we present the Hartigans' Dip Test results. In Figure 6, we first show the distributions of raw scores (x) across all problems in the task, if they are evaluated using F-1, ROUGE, or EditSim. We can easily tell that some tasks exhibit clear bimodality, most notably BAMBOO SenHallu and AbsHallu, in which cases accuracy might also be used instead. Many other QA tasks, including HotpotQA, 2WikiMultihopQA, and TriviaQA, also have the most probability mass at 0 and 1, which suggests that the majority of the problems at least should be categorized into the COW scenario. We also found in these cases, 0.5 (or 50 percentage) is generally a reasonable threshold to set apart "1" from "0". On the other hand, most summarization tasks have a unimodal probability distribution and little probability mass at 0 or 1.

We first bucketize the scores (between 0 and 1) from all observations with an equal width of 0.1, and then apply the Hartigans' Dip Test to the bucketized scores for each task. We calculate the p-value for each task, and show $1-$ p-value in Figure 7. We assign a small value to the tasks when their y-axis is 0. A short bar (high p-value) indicates more likely a PIG scenario (unimodal), and a tall bar (low p-value) indicates more likely a COW scenario (multi-modal). We use ∗ to suffix a task name and a blue bar when it is evaluated using F-1, † and an orange bar when it it evaluated using ROUGE, and ◇ and a grey bar when it is evaluated using EditSim. We see while most ROUGE-evaluated tasks belong to the PIG scenario and most F1-evaluated tasks belong to the COW scenario, there also exist several exceptions.

We found that both COW and PIG scenarios can co-exist among the problems within the same task. Next, we apply the Hartigans' Dip Test at each problem level. We show the result in Figure 7. The x-axis is $1-$ p-value and the y-axis is the number of problems. A bar at $x = 0$ indicates more likely a PIG scenario, and a bar at $x = 1$ indicates more likely a COW scenario. We split the problem set of each task into a COW subset and a PIG subset, if they both have more than ten problems. Otherwise, we treat the problem set entirely as COW or PIG without subsetting. We report the final decision in Table 5. We note that all tasks that are evaluated using accuracy are considered COW only.

Table 5: Problem level Hartigans' Dip Test results for the tasks that are evaluated using F-1, ROUGE, or EditSim.

| MIXED COW & PIG | COW ONLY | PIG ONLY |
| --- | --- | --- |
| L-Eval MultiDoc2Dial | L-Eval NQ | L-Eval MultiNews |
| L-Eval Qasper | BAMBOO SenHallu 4K / 16K | L-Eval GovReport |
| L-Eval LongFQA | BAMBOO AbsHallu 4K / 16K | L-Eval BigPatent |
| L-Eval CUAD | LongBench Trivia QA | L-Eval SummScreen |
| L-Eval NarrativeQA | | L-Eval OpenReview |
| LongBench NarrativeQA | | L-Eval QMSum |
| LongBench Qasper | | L-Eval SPACE |
| LongBench MultiFieldQA | | LongBench GovReport |
| LongBench HotpotQA | | LongBench QMSum |
| LongBench 2WikiMultihopQA | | LongBench MultiNews |
| LongBench MuSiQue | | |
| LongBench PrivateEval 4K / 16K | | |
| LongBench SamSum | | |
| LongBench LCC | | |
| LongBench RepoBench-p | | |

## I  TASKS RANKED BY CATEGORIES III & V

We replot Figure 2 into Figures 9 and 10, where the tasks are sorted from the most retrieval focused (top) to the least (bottom) in (a) and from the most holistic understanding focused (top) to the least (bottom) in (b). **Categories I** and **II** are stacked to the left of the vertical axis, and **III** to **V** are stacked to the right.

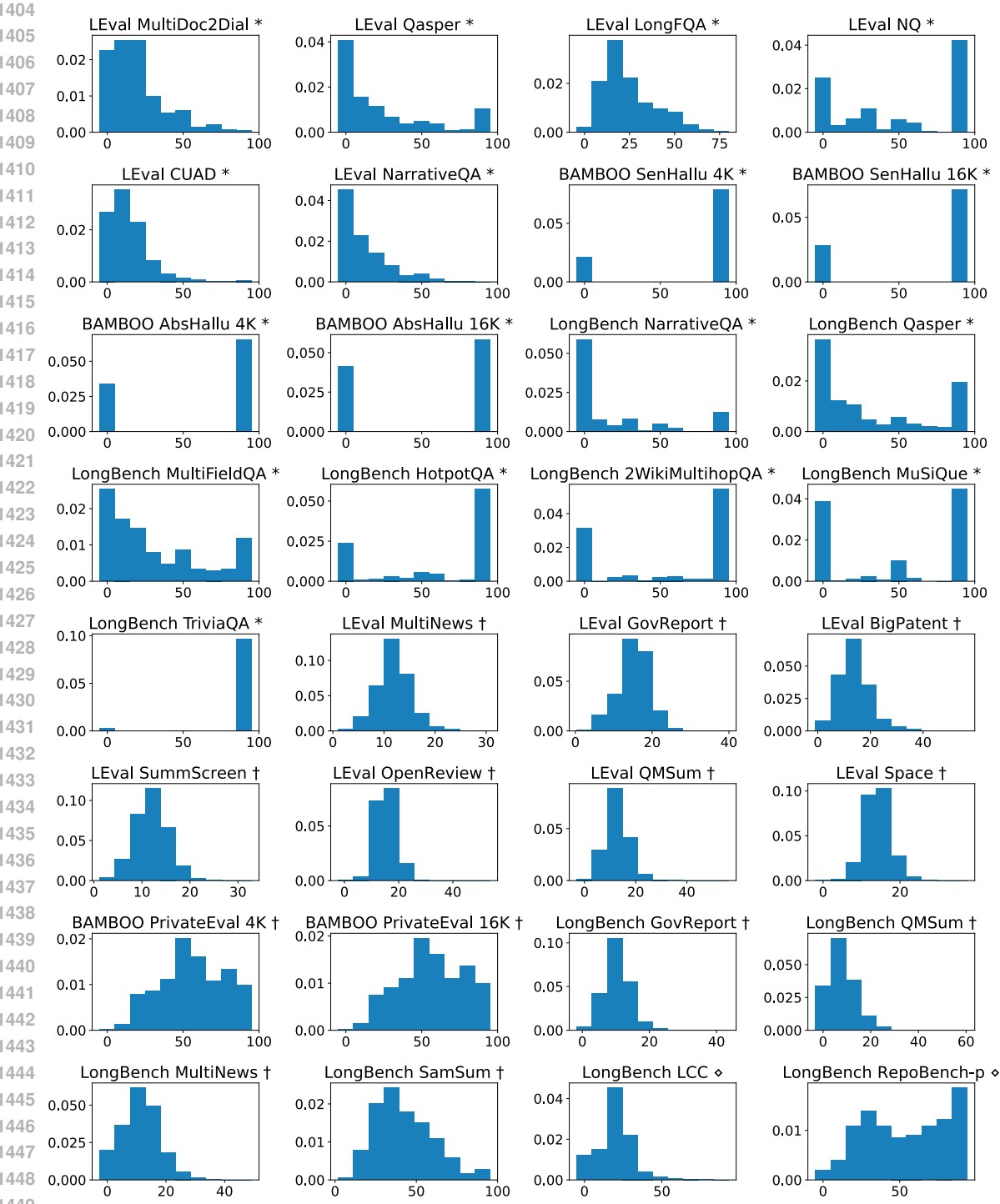

Figure 6: Raw score (x) distribution of all problems in each task that is evaluated using F-1, ROUGE, or EditSim. We use ∗ to suffix a task name when it is evaluated using F-1, † when it it evaluated using ROUGE, and ⋄ when it is evaluated using EditSim. The x-axis is the percentage score ($100s$) and the y-axis is the probability mass.

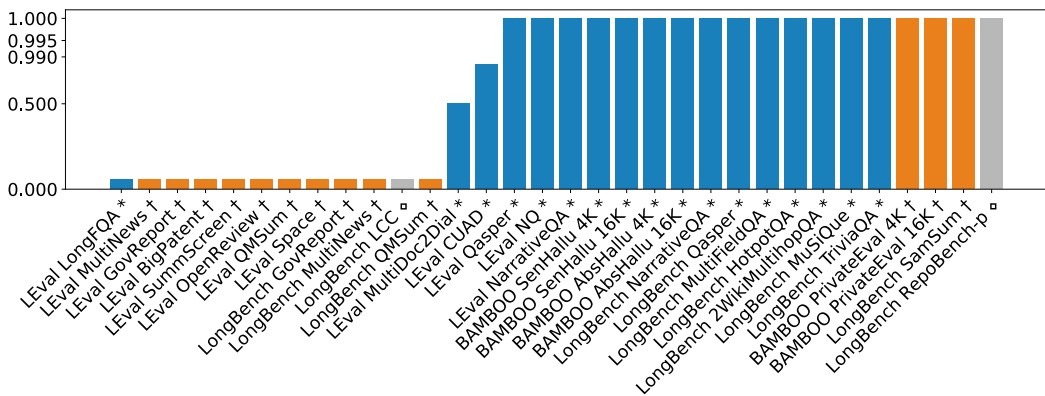

Figure 7: Hartigans' Dip Test results at the task level. We use ∗ to suffix a task name and a blue bar when it is evaluated using F-1, † and an orange bar when it it evaluated using ROUGE, and ⋄ and a grey bar when it is evaluated using EditSim. The y-axis is $1-$ p-value, scaled using $\sinh$. We assign a small value to the tasks when their y-axis is 0. A short bar indicates more likely a PIG scenario, and a tall bar indicates more likely a COW scenario.

## J  OUTCOMES & FULL PARAMETERS OF REPRESENTATIVE PROBLEMS IN QUALITY & LONGFQA

In this section, we again focus on the tasks QuALITY and LongFQA, and we provide additional information beyond Table 2. We first report $\lambda$ and $k$ estimated by our proposed method before applying the category assignment step (using the thresholds $\lambda_p$, $k_p$, and $\lambda_q$) in Figure 11, and then we show the detailed results for the most representative problems, which include original outcomes $P(\mathrm{x})$, the membership probability $q(\mathrm{z} = \mathcal{O}|\mathrm{x} = x)$, hybrid probability for the oracle component $P(\mathrm{x} = x|\mathcal{O})$. as well as the background noise component $P(\mathrm{x} = x|\mathcal{N})$, and $P(\mathrm{z} = z)$.

### J.1  QUALITY

#### MOST REPRESENTATIVE QUESTION FOR CATEGORY II

Why might one not want to live in the universe in which this story takes place?

#### GROUND TRUTH ANSWER

(C) Survival itself is difficult

#### OUTCOMES & LENGTH-SPECIFIC PARAMETERS

| C | $P(\mathrm{x} = \cdot)$ | | | $q(\mathrm{z} = \mathcal{O}|\mathrm{x} = \cdot)$ | | | $P(\mathrm{x} = \cdot|\mathcal{O})$ | | |
|---|---|---|---|---|---|---|---|---|---|
| | 1 | 0 | $\emptyset$ | 1 | 0 | $\emptyset$ | 1 | 0 | $\emptyset$ |
| 0 | 0.00 | 0.00 | 1.00 | 0.00 | 0.00 | 0.97 | 0.00 | 0.00 | 1.00 |
| 1 | 0.23 | 0.00 | 0.76 | 1.00 | 0.00 | 0.96 | 0.20 | 0.00 | 0.80 |
| 2 | 0.43 | 0.00 | 0.56 | 1.00 | 0.00 | 0.95 | 0.36 | 0.00 | 0.64 |
| 5 | 0.68 | 0.01 | 0.31 | 1.00 | 0.00 | 0.90 | 0.68 | 0.00 | 0.32 |
| 10 | 0.82 | 0.03 | 0.15 | 1.00 | 0.00 | 0.75 | 0.90 | 0.00 | 0.10 |
| 20 | 0.92 | 0.01 | 0.06 | 1.00 | 0.00 | 0.23 | 0.99 | 0.00 | 0.01 |
| 50 | 1.00 | 0.00 | 0.00 | 1.00 | 0.00 | 0.00 | 1.00 | 0.00 | 0.00 |
| 100 | 1.00 | 0.00 | 0.00 | 1.00 | 0.00 | 0.00 | 1.00 | 0.00 | 0.00 |
| 498 | 1.00 | 0.00 | 0.00 | 1.00 | 0.00 | 0.00 | 1.00 | 0.00 | 0.00 |

#### TASK OR PROBLEM-SPECIFIC PARAMETERS

| $\lambda$ | $k$ | $P(x = \cdot \mid z = \mathcal{N})$ | | | $P(z = \cdot)$ | |
|---|---|---|---|---|---|---|
| | | 1 | 0 | $\emptyset$ | $\mathcal{O}$ | $\mathcal{N}$ |
| 1 | 100 | 0.01 | 0.05 | 0.94 | 0.96 | 0.04 |

### MOST REPRESENTATIVE QUESTION FOR CATEGORY III

Why does the text mean when it says that Korvin was "unconscious" at the time of his lessons in the local language?

### GROUND TRUTH ANSWER

(A) It means that the Tr'en put Korvin under drug hypnosis while they taught him their language.

### OUTCOMES & LENGTH-SPECIFIC PARAMETERS

| C | $P(x = \cdot)$ | | | $q(z = \mathcal{O} \mid x = \cdot)$ | | | $P(x = \cdot \mid \mathcal{O})$ | | |
|---|---|---|---|---|---|---|---|---|---|
| | 1 | 0 | $\emptyset$ | 1 | 0 | $\emptyset$ | 1 | 0 | $\emptyset$ |
| 0 | 0.00 | 0.00 | 1.00 | 0.00 | 0.00 | 0.60 | 0.00 | 0.00 | 1.00 |
| 1 | 0.00 | 0.00 | 0.99 | 0.26 | 0.00 | 0.60 | 0.00 | 0.00 | 1.00 |
| 2 | 0.00 | 0.00 | 0.99 | 0.41 | 0.00 | 0.60 | 0.00 | 0.00 | 1.00 |
| 5 | 0.01 | 0.01 | 0.98 | 0.64 | 0.00 | 0.60 | 0.01 | 0.00 | 0.99 |
| 10 | 0.02 | 0.03 | 0.95 | 0.78 | 0.00 | 0.60 | 0.02 | 0.00 | 0.98 |
| 20 | 0.05 | 0.04 | 0.91 | 0.88 | 0.00 | 0.59 | 0.05 | 0.00 | 0.95 |
| 50 | 0.13 | 0.01 | 0.87 | 0.95 | 0.00 | 0.57 | 0.12 | 0.00 | 0.88 |
| 100 | 0.15 | 0.00 | 0.85 | 0.97 | 0.00 | 0.53 | 0.25 | 0.00 | 0.75 |
| 408 | 1.00 | 0.00 | 0.00 | 0.99 | 0.00 | 0.00 | 1.00 | 0.00 | 0.00 |

### TASK OR PROBLEM-SPECIFIC PARAMETERS

| $\lambda$ | $k$ | $P(x = \cdot \mid z = \mathcal{N})$ | | | $P(z = \cdot)$ | |
|---|---|---|---|---|---|---|
| | | 1 | 0 | $\emptyset$ | $\mathcal{O}$ | $\mathcal{N}$ |
| 1 | 1 | 0.01 | 0.05 | 0.94 | 0.59 | 0.41 |

### MOST REPRESENTATIVE QUESTION FOR CATEGORY IV

Why would Tom Dorr frame Asa Graybar for stealing the Slider egg?

### GROUND TRUTH ANSWER

(A) Graybar's discoveries could ruin the Hazeltyne business.

### OUTCOMES & LENGTH-SPECIFIC PARAMETERS

| C | $P(x = \cdot)$ | | | $q(z = \mathcal{O} \mid x = \cdot)$ | | | $P(x = \cdot \mid \mathcal{O})$ | | |
|---|---|---|---|---|---|---|---|---|---|
| | 1 | 0 | $\emptyset$ | 1 | 0 | $\emptyset$ | 1 | 0 | $\emptyset$ |
| 0 | 0.00 | 0.00 | 1.00 | 0.00 | 0.00 | 0.65 | 0.00 | 0.00 | 1.00 |
| 1 | 0.00 | 0.00 | 1.00 | 0.00 | 0.00 | 0.65 | 0.00 | 0.00 | 1.00 |
| 2 | 0.01 | 0.00 | 0.99 | 0.00 | 0.00 | 0.65 | 0.00 | 0.00 | 1.00 |
| 5 | 0.00 | 0.00 | 1.00 | 0.00 | 0.00 | 0.65 | 0.00 | 0.00 | 1.00 |
| 10 | 0.00 | 0.00 | 1.00 | 0.00 | 0.00 | 0.65 | 0.00 | 0.00 | 1.00 |
| 20 | 0.01 | 0.00 | 0.99 | 0.00 | 0.00 | 0.65 | 0.00 | 0.00 | 1.00 |

| 50 | 0.03 | 0.00 | 0.97 | 0.35 | 0.00 | 0.65 | 0.00 | 0.00 | 1.00 |
| 100 | 0.05 | 0.00 | 0.95 | 0.96 | 0.00 | 0.61 | 0.15 | 0.00 | 0.85 |
| 381 | 1.00 | 0.00 | 0.00 | 0.99 | 0.00 | 0.00 | 1.00 | 0.00 | 0.00 |

### TASK OR PROBLEM-SPECIFIC PARAMETERS

| $\lambda$ | $k$ | $P(\mathrm{x} = \cdot \mid z = \mathcal{N})$ | | | $P(z = \cdot)$ | |
| | | 1 | 0 | $\emptyset$ | $\mathcal{O}$ | $\mathcal{N}$ |
| --- | --- | --- | --- | --- | --- | --- |
| 50 | 1 | 0.01 | 0.05 | 0.94 | 0.64 | 0.36 |

### MOST REPRESENTATIVE QUESTION FOR CATEGORY V

How many sentences does this story have approximately?

### GROUND TRUTH ANSWER

(D) 406

### OUTCOMES & LENGTH-SPECIFIC PARAMETERS

| C | $P(\mathrm{x} = \cdot)$ | | | $q(z = \mathcal{O} \mid \mathrm{x} = \cdot)$ | | | $P(\mathrm{x} = \cdot \mid \mathcal{O})$ | | |
| | 1 | 0 | $\emptyset$ | 1 | 0 | $\emptyset$ | 1 | 0 | $\emptyset$ |
| --- | --- | --- | --- | --- | --- | --- | --- | --- | --- |
| 0 | 0.00 | 0.00 | 1.00 | 0.00 | 0.00 | 0.69 | 0.00 | 0.00 | 1.00 |
| 1 | 0.00 | 0.00 | 1.00 | 0.00 | 0.00 | 0.69 | 0.00 | 0.00 | 1.00 |
| 2 | 0.00 | 0.00 | 1.00 | 0.00 | 0.00 | 0.69 | 0.00 | 0.00 | 1.00 |
| 5 | 0.00 | 0.00 | 1.00 | 0.00 | 0.00 | 0.69 | 0.00 | 0.00 | 1.00 |
| 10 | 0.00 | 0.00 | 1.00 | 0.00 | 0.00 | 0.69 | 0.00 | 0.00 | 1.00 |
| 20 | 0.00 | 0.00 | 1.00 | 0.00 | 0.00 | 0.69 | 0.00 | 0.00 | 1.00 |
| 50 | 0.00 | 0.00 | 1.00 | 0.00 | 0.00 | 0.69 | 0.00 | 0.00 | 1.00 |
| 100 | 0.00 | 0.01 | 0.99 | 0.00 | 0.00 | 0.69 | 0.00 | 0.00 | 1.00 |
| 408 | 0.00 | 0.00 | 1.00 | 1.00 | 0.00 | 0.00 | 1.00 | 0.00 | 0.00 |

### TASK OR PROBLEM-SPECIFIC PARAMETERS

| $\lambda$ | $k$ | $P(\mathrm{x} = \cdot \mid z = \mathcal{N})$ | | | $P(z = \cdot)$ | |
| | | 1 | 0 | $\emptyset$ | $\mathcal{O}$ | $\mathcal{N}$ |
| --- | --- | --- | --- | --- | --- | --- |
| $L$ | 1 | 0.01 | 0.05 | 0.94 | 0.68 | 0.32 |

## J.2 LONGFQA

### MOST REPRESENTATIVE QUESTION FOR CATEGORY II

What are some key accomplishments of FS KKR Capital Corp. in 2018 as mentioned in the call?

### GROUND TRUTH ANSWER

Key accomplishments included receiving shareholder approval for the partnership between FS Investments and KKR, optimizing the company's capital structure by closing a $2.1 billion revolver, completing a merger between CCT and FSIC, and starting to capitalize on the benefits of the combined FS Investments and KKR platforms.

### OUTCOMES & LENGTH-SPECIFIC PARAMETERS

| C | $P(x = \cdot)$ | | | $q(z = \mathcal{O}|x = \cdot)$ | | | $P(x = \cdot|\mathcal{O})$ | | |
|---|---|---|---|---|---|---|---|---|---|
| | 1 | 0 | $\emptyset$ | 1 | 0 | $\emptyset$ | 1 | 0 | $\emptyset$ |
| 0 | 0.00 | 0.00 | 1.00 | 0.00 | 0.00 | 0.67 | 0.00 | 0.00 | 1.00 |
| 1 | 0.00 | 0.22 | 0.78 | 0.00 | 0.00 | 0.67 | 0.00 | 0.00 | 1.00 |
| 2 | 0.01 | 0.27 | 0.72 | 0.00 | 0.00 | 0.67 | 0.00 | 0.00 | 1.00 |
| 5 | 0.04 | 0.57 | 0.39 | 0.84 | 0.00 | 0.65 | 0.09 | 0.00 | 0.91 |
| 10 | 0.07 | 0.71 | 0.22 | 0.97 | 0.00 | 0.51 | 0.51 | 0.00 | 0.49 |
| 20 | 0.19 | 0.80 | 0.01 | 0.98 | 0.00 | 0.18 | 0.89 | 0.00 | 0.11 |
| 50 | 0.37 | 0.63 | 0.00 | 0.98 | 0.00 | 0.00 | 1.00 | 0.00 | 0.00 |
| 100 | 1.00 | 0.00 | 0.00 | 0.98 | 0.00 | 0.00 | 1.00 | 0.00 | 0.00 |
| 119 | 1.00 | 0.00 | 0.00 | 0.98 | 0.00 | 0.00 | 1.00 | 0.00 | 0.00 |

## TASK OR PROBLEM-SPECIFIC PARAMETERS

| $\lambda$ | $k$ | $P(x = \cdot|z = \mathcal{N})$ | | | $P(z = \cdot)$ | |
|---|---|---|---|---|---|---|
| | | 1 | 0 | $\emptyset$ | $\mathcal{O}$ | $\mathcal{N}$ |
| 5 | 10 | 0.01 | 0.73 | 0.26 | 0.35 | 0.65 |

## MOST REPRESENTATIVE QUESTION FOR CATEGORY III

What were the consolidated revenue and the revenue growth for the Surgical Product segment over last year?

## GROUND TRUTH ANSWER

The consolidated revenue for the company was $21.6 million, which is our highest quarterly revenue ever and up 7% over last year's first quarter. The revenue for the Surgical Product segment was $15.5 million, which is also our highest quarterly surgical products revenue ever, and was up 8% over last year.

## OUTCOMES & LENGTH-SPECIFIC PARAMETERS

| C | $P(x = \cdot)$ | | | $q(z = \mathcal{O}|x = \cdot)$ | | | $P(x = \cdot|\mathcal{O})$ | | |
|---|---|---|---|---|---|---|---|---|---|
| | 1 | 0 | $\emptyset$ | 1 | 0 | $\emptyset$ | 1 | 0 | $\emptyset$ |
| 0 | 0.00 | 0.00 | 1.00 | 0.00 | 0.00 | 0.92 | 0.00 | 0.00 | 1.00 |
| 1 | 0.00 | 0.01 | 0.99 | 0.00 | 0.00 | 0.92 | 0.00 | 0.00 | 1.00 |
| 2 | 0.01 | 0.02 | 0.97 | 0.68 | 0.00 | 0.92 | 0.01 | 0.00 | 0.99 |
| 5 | 0.03 | 0.03 | 0.93 | 0.89 | 0.00 | 0.92 | 0.03 | 0.00 | 0.97 |
| 10 | 0.06 | 0.07 | 0.86 | 0.95 | 0.00 | 0.92 | 0.06 | 0.00 | 0.94 |
| 20 | 0.13 | 0.18 | 0.69 | 0.98 | 0.00 | 0.91 | 0.12 | 0.00 | 0.88 |
| 50 | 0.12 | 0.53 | 0.35 | 0.99 | 0.00 | 0.89 | 0.31 | 0.00 | 0.69 |
| 100 | 0.12 | 0.88 | 0.00 | 1.00 | 0.00 | 0.81 | 0.63 | 0.00 | 0.37 |
| 157 | 1.00 | 0.00 | 0.00 | 1.00 | 0.00 | 0.00 | 1.00 | 0.00 | 0.00 |

## TASK OR PROBLEM-SPECIFIC PARAMETERS

| $\lambda$ | $k$ | $P(x = \cdot|z = \mathcal{N})$ | | | $P(z = \cdot)$ | |
|---|---|---|---|---|---|---|
| | | 1 | 0 | $\emptyset$ | $\mathcal{O}$ | $\mathcal{N}$ |
| 2 | 1 | 0.01 | 0.73 | 0.26 | 0.76 | 0.24 |

## MOST REPRESENTATIVE QUESTION FOR CATEGORY IV

What impact did the introduction of the Valved Tearaway and Pediatric Microslide Introducer products have on the company's market position and potential future sales?

### GROUND TRUTH ANSWER

The Valved Tearaway product, which gained FDA clearance, is intended to compete with a significant market leader. Initial interest in the product was brisk, with estimated sales of about $300,000 in its first year and in excess of $1 million annually in subsequent years. The Pediatric Microslide Introducer, developed in response to requests from pediatric nurses, may not itself be a massive revenue generator, but it has enabled the company to gain access to accounts previously unavailable for other vascular products. This has positioned the company as a listener and problem-solver within the industry, enhancing our reputation and possibly future sales.

### OUTCOMES & LENGTH-SPECIFIC PARAMETERS

| C | $P(x = \cdot)$ | | | $q(z = \mathcal{O}|x = \cdot)$ | | | $P(x = \cdot|\mathcal{O})$ | | |
|---|---|---|---|---|---|---|---|---|---|
| | 1 | 0 | $\emptyset$ | 1 | 0 | $\emptyset$ | 1 | 0 | $\emptyset$ |
| 0 | 0.00 | 1.00 | 0.00 | 0.00 | 0.01 | 0.82 | 0.00 | 0.00 | 1.00 |
| 1 | 0.00 | 0.39 | 0.61 | 0.00 | 0.01 | 0.82 | 0.00 | 0.00 | 1.00 |
| 2 | 0.00 | 0.25 | 0.75 | 0.00 | 0.01 | 0.82 | 0.00 | 0.00 | 1.00 |
| 5 | 0.00 | 0.14 | 0.86 | 0.00 | 0.01 | 0.82 | 0.00 | 0.00 | 1.00 |
| 10 | 0.01 | 0.16 | 0.83 | 0.00 | 0.01 | 0.82 | 0.00 | 0.00 | 1.00 |
| 20 | 0.01 | 0.28 | 0.71 | 0.49 | 0.00 | 0.82 | 0.01 | 0.00 | 0.99 |
| 50 | 0.08 | 0.46 | 0.45 | 0.97 | 0.00 | 0.78 | 0.22 | 0.00 | 0.78 |
| 100 | 0.14 | 0.86 | 0.00 | 0.99 | 0.00 | 0.66 | 0.59 | 0.00 | 0.41 |
| 157 | 0.00 | 1.00 | 0.00 | 0.99 | 0.00 | 0.00 | 1.00 | 0.00 | 0.00 |

### TASK OR PROBLEM-SPECIFIC PARAMETERS

| $\lambda$ | $k$ | $P(x = \cdot|z = \mathcal{N})$ | | | $P(z = \cdot)$ | |
|---|---|---|---|---|---|---|
| | | 1 | 0 | $\emptyset$ | $\mathcal{O}$ | $\mathcal{N}$ |
| 20 | 1 | 0.01 | 0.73 | 0.26 | 0.55 | 0.45 |

### MOST REPRESENTATIVE QUESTION FOR CATEGORY V

Does JLL have greater market share in U.S. leasing than in Capital Markets?

### GROUND TRUTH ANSWER

Yes, much greater. This is our powerhouse, the U.S. leasing and tenant rep business, and it continues to grow much stronger than the market is offering.

### OUTCOMES & LENGTH-SPECIFIC PARAMETERS

| C | $P(x = \cdot)$ | | | $q(z = \mathcal{O}|x = \cdot)$ | | | $P(x = \cdot|\mathcal{O})$ | | |
|---|---|---|---|---|---|---|---|---|---|
| | 1 | 0 | $\emptyset$ | 1 | 0 | $\emptyset$ | 1 | 0 | $\emptyset$ |
| 0 | 0.00 | 0.00 | 1.00 | 0.00 | 0.00 | 1.00 | 0.00 | 0.00 | 1.00 |
| 1 | 0.00 | 0.00 | 1.00 | 0.00 | 0.00 | 1.00 | 0.00 | 0.00 | 1.00 |
| 2 | 0.00 | 0.00 | 1.00 | 0.00 | 0.00 | 1.00 | 0.00 | 0.00 | 1.00 |
| 5 | 0.00 | 0.00 | 1.00 | 0.00 | 0.00 | 1.00 | 0.00 | 0.00 | 1.00 |
| 10 | 0.00 | 0.00 | 1.00 | 0.00 | 0.00 | 1.00 | 0.00 | 0.00 | 1.00 |
| 20 | 0.00 | 0.00 | 1.00 | 0.00 | 0.00 | 1.00 | 0.00 | 0.00 | 1.00 |
| 50 | 0.00 | 0.00 | 1.00 | 0.00 | 0.00 | 1.00 | 0.00 | 0.00 | 1.00 |
| 100 | 0.00 | 0.00 | 1.00 | 0.00 | 0.00 | 1.00 | 0.00 | 0.00 | 1.00 |
| 162 | 0.00 | 0.00 | 1.00 | 1.00 | 0.00 | 0.00 | 1.00 | 0.00 | 0.00 |

TASK OR PROBLEM-SPECIFIC PARAMETERS

| $\lambda$ | $k$ | $P(\mathrm{x} = \cdot|z = \mathcal{N})$ | | | $P(\mathrm{z} = \cdot)$ | |
|---|---|---|---|---|---|---|
| | | 1 | 0 | $\emptyset$ | $\mathcal{O}$ | $\mathcal{N}$ |
| $L$ | 1 | 0.01 | 0.73 | 0.26 | 1.00 | 0.00 |

## K  FURTHER ANALYSIS

### K.1  SAMPLING STRATEGIES

We compare between a few simple heuristics, including sampling at a fixed rate $r$ (**sample rate**), taking every $N$-th (fixed) candidate in the sequence (**take every**), sampling inversely proportionally to the observation length (i.e. sampling more for short observations and less for long observations, **sample ip rate**), and taking every $N$-th (inversely proportionally to the observation length) candidate (**take ip rate**), as well as observing only long observations (**>= 10 only**), short observations (**<= 10 only**), and wider observation length intervals (**0, 1, 5, 20, 100, max only**).

We plot the Relative Change ($\delta$) and Spearman's rank correlation coefficient ($\rho$) of $\lambda$ and $k$ as well as the KL Divergence of $P(\mathrm{x}|z = \mathcal{N})$ between enumerating all observation spans and various sampling strategies for L-Eval TOEFL and SFcition in Figures 12 and 13 respectively. To quantify the required resources (i.e. the x-axis), we use the number of total units and the number of total tokens as proxies. First, we found that the more the samples the more highly the inferred parameters can correlate with those estimated from using all the possible observation spans. Then, we found that "take" strategies (i.e. shifting the observation window by a fixed number of units) work better than the random "sampling" strategies, suggesting that continuity exists in the sequence, i.e. the correctness of a span is positively correlated with that of its neighboring span. Also, the take-every strategy generally works well. For example, in the case of L-Eval SFcition, using take-every-5 strategy (i.e. reducing the total required resource by 80%), we can still obtain $\rho(\lambda)$ of 0.93, $\rho(k)$ of 0.99, and KL Divergence of $P(\mathrm{x}|z = \mathcal{N})$ of $3.7 \times 10^{-5}$.

### K.2  UNIT GRANULARITIES

We use different unit granularities for seven tasks among the benchmark suites: L-Eval CodeU, BAM-BOO PrivateEval 4K/16K, LongBench HotpotQA, 2WikiMultiHopQA, MuSiQue, and MultiNews. Detailed preprocessing specs for these tasks can be found in Table 3, where each task has two rows corresponding to two unit granularity options. We report the Relative Change ($\delta$) and Spearman's rank correlation coefficient ($\rho$) of $\lambda$ and $k$ and KL Divergence of $P(X|Z = \mathcal{N})$ between different unit granularity selections for the same tasks in Table 8. With the unit definition change, the obtained $\lambda$ now measures the length of the span (in the COW scenario) or the number of length-1 aspects (in the PIG scenario) w.r.t the new unit. Therefore, we note that $\delta(\lambda)$ might not be a reliable measure for $\lambda$, and instead $\rho$, which ignores the exact number and uses the rank among the peers, is more preferred. When the model predicts the same $k$ for all the problems (most likely $k = 1$), the Spearman's rank correlation coefficient $\rho(k)$ becomes undefined (NaN). We see in these cases, $\delta(k)$ is often 0 or close to 0, which we may also interpret as high similarity.

We see that in the COW scenario, the ranking of $\lambda$ and the ranking of $k$ are both preserved between different unit granularities, with $\rho(\lambda)$ between 0.54 and 0.80 and $\rho(\lambda) \geq 0.50$, i.e. strong correlation. The ranking of $\lambda$ is sometimes less preserved in the PIG scenario, with $\rho(\lambda)$ between 0.35 and 0.84 across the tasks, corresponding to moderate to strong correlation, while the ranking of $k$ is also well preserved with $\rho(k) \geq 0.64$. The background noise distribution estimation is mostly preserved as well, with the KL divergence $\leq 0.18$ across all tasks and subsets.

We found several reasons to explain the minor discordance between the parameter rankings derived from the two unit granularities. First, it can sometimes be attributed to the rather noticeable difference within $P_{\mathrm{nonpar}}(\mathrm{x}|z = \mathcal{O})$ as part of the hybrid oracle component. We recall that we have assumed this parameter is shared only among the observations of a single problem, instead of all observations of

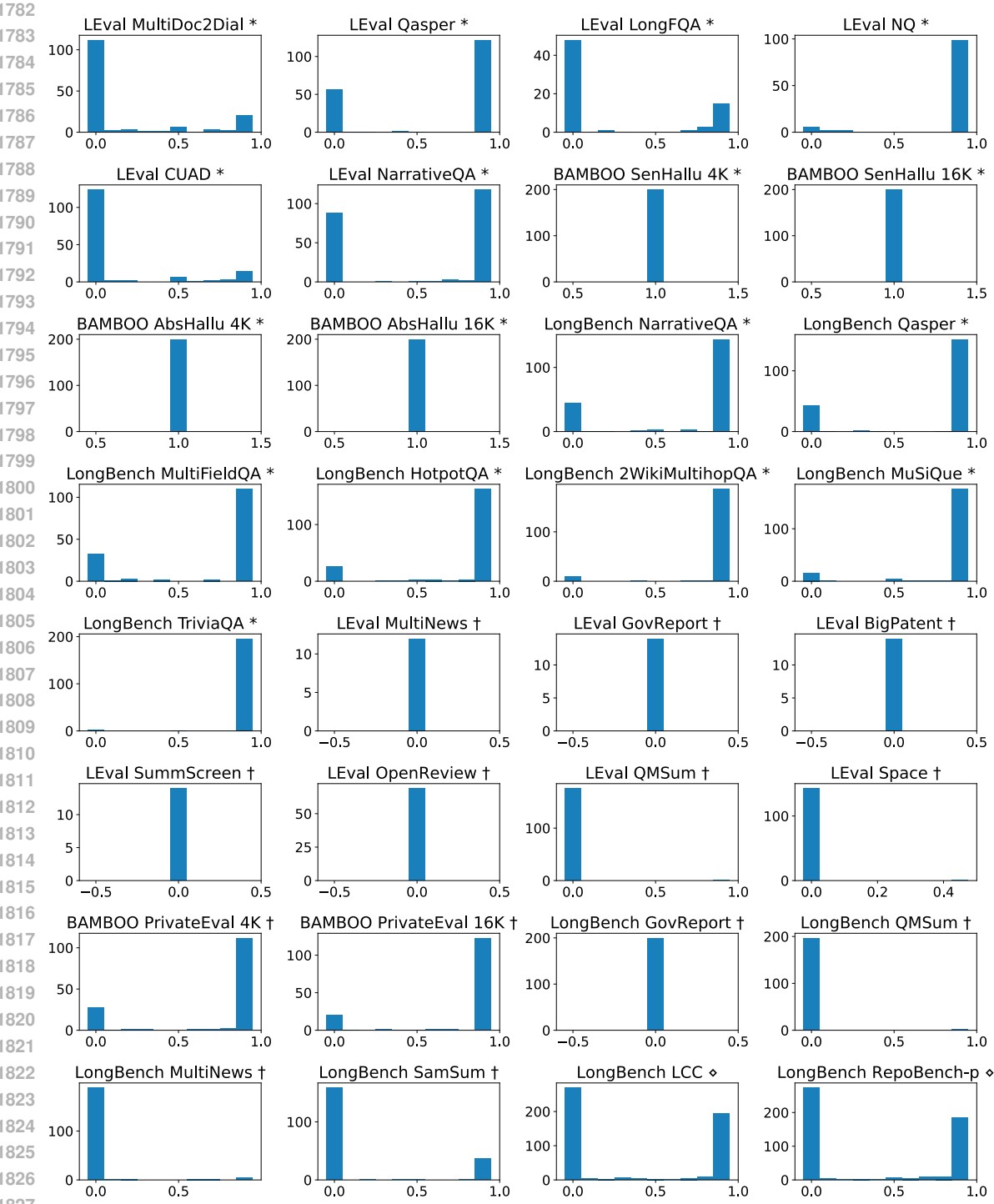

Figure 8: Hartigans' Dip Test results at the problem level. We use ∗ to suffix a task name when it is evaluated using F-1, † when it it evaluated using ROUGE, and ⋄ when it is evaluated using EditSim. The x-axis is $1-$ p-value and the y-axis is the number of problems. A bar at $x = 0$ indicates more likely a PIG scenario, and a bar at $x = 1$ indicates more likely a COW scenario.

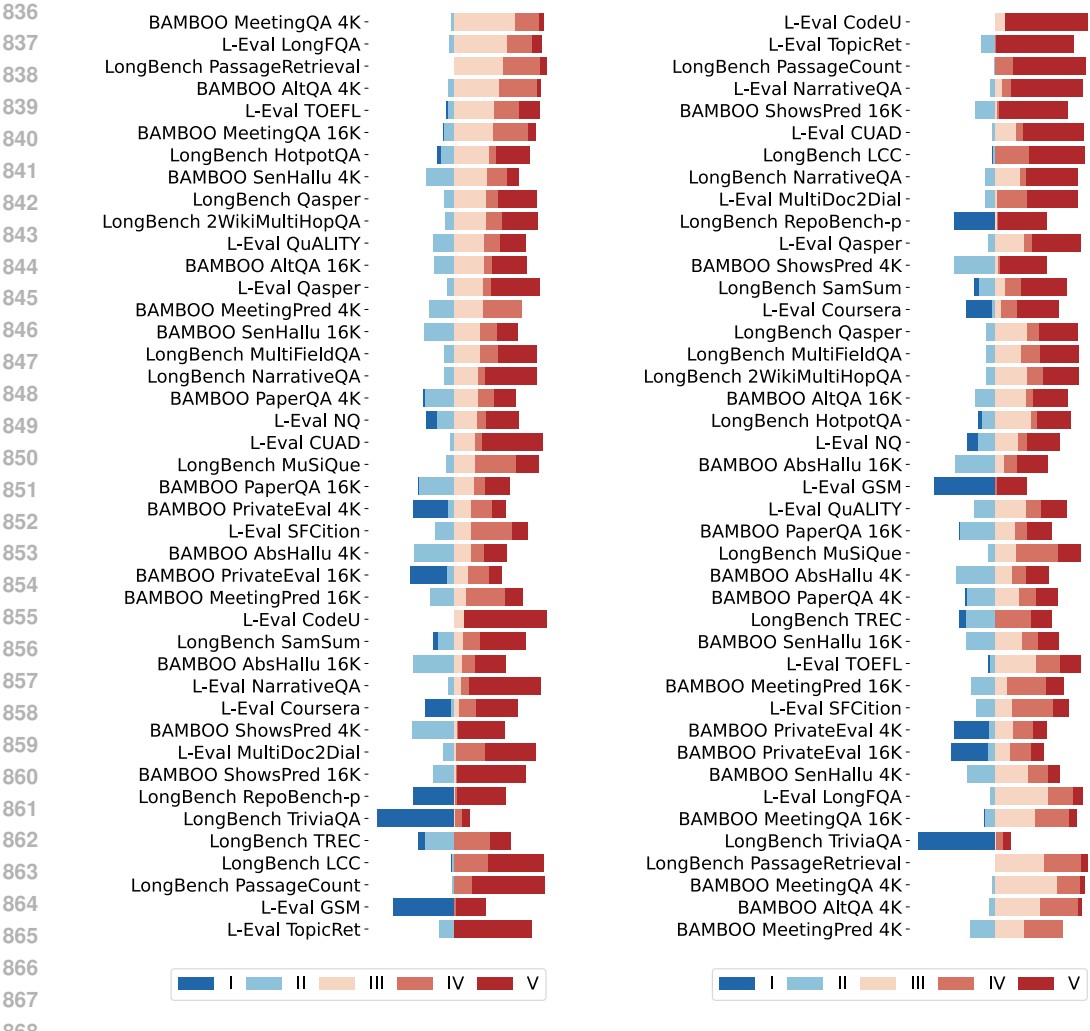

(a) From most retrieval focused (top) to least retrieval focused (bottom).

(b) From most holistic understanding focused (top) to least holistic understanding focused (bottom).

Figure 9: Category distribution of problems in each task using the COW assumption. Bars are aligned such that **Category I** and **II** are shown on the left and **Category III** to **V** are shown on the right.

all problems for the task, and computed using only observations whose $C < \lambda$, which can be a rather small set and thus sensitive to the outcomes observed for small $C$. A hierarchical assumption that a problem-specific $P_{\text{nonpar}}(x|z = \mathcal{O})$ is generated from a task-specific meta distribution might help alleviate the issue.

Second, we found that this also happens when there is huge discrepancy between the context lengths when measured with the two unit granularities. Consider an example where problem $i$ has a ground-truth span that contains 4 paragraphs, with each paragraph having 5 sentences, and problem $j$ has a ground-truth span that contains 8 paragraphs, with each paragraph having 2 sentences. If we use paragraphs as units, then we have $\lambda_i = 4 < 8 = \lambda_j$. If we use sentences as units, then we have $\lambda_i = 20 > 16 = \lambda_j$. In this case, we should use a unit granularity with which the context lengths have little variance when measured in tokens.

Third, in the PIG scenario, this might also be due to the fact that some aspects that are scattered across multiple small granular spans may occur far from each other and thus require the same number of larger granular spans, while others may occur close to each other and require much fewer larger granular spans. We believe this is an expected outcome, as we do not explicitly model the locations of the ground-truth spans.

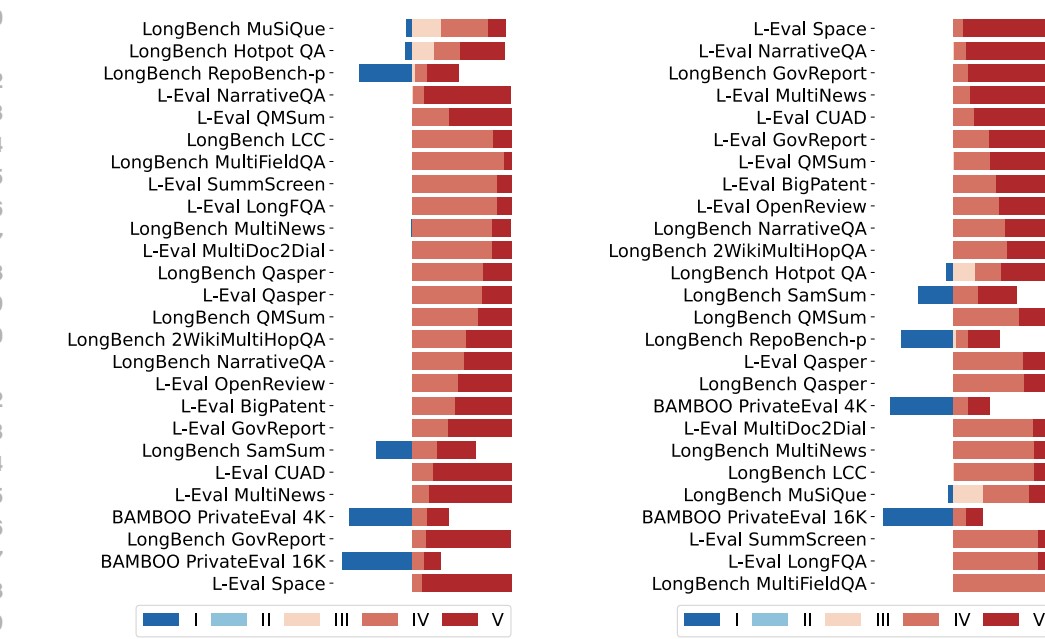

(a) From most retrieval focused (top) to least retrieval focused (bottom).

(b) From most holistic understanding focused (top) to least holistic understanding focused (bottom).

Figure 10: Category distribution of problems in each task using the PIG assumption. Bars are aligned such that **Category I** and **II** are shown on the left and **Category III** to **V** are shown on the right.

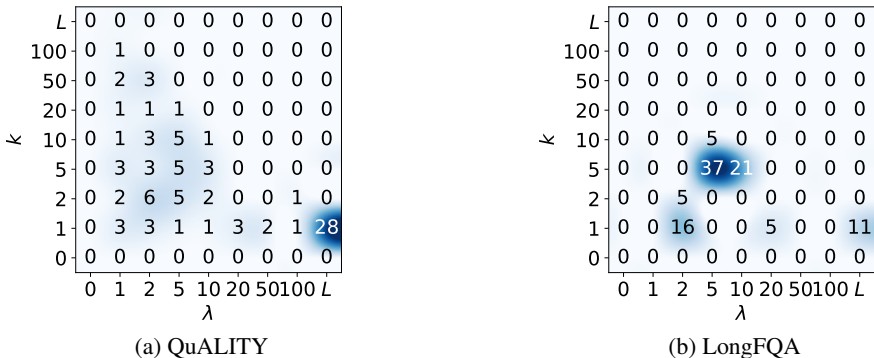

(a) QuALITY

(b) LongFQA

Figure 11: Focus category distribution (in percentage) on the two-dimensional $\lambda$-$k$ plane.

### K.3 PROBING MODELS: GEMINI 1.5 FLASH VS PALM 2-S

Throughout the paper, we report the results using the Gemini 1.5 Flash model as the probing model. In this section, we compare with another independently trained model PaLM 2-S (Anil et al., 2023). We note that they differ in the model architecture as well as the training data mixture. This changes not only the CBZS pattern, but also the short context understanding capability. We omit the problems who are assigned to **Category I** (CBZS) by either model.

We show the Relative Change ($\delta$) and the Spearman's rank correlation coefficient ($\rho$) of $\lambda$ and $k$ between the Gemini 1.5 Flash model and the PaLM 2-S model across the tasks in Figure 14. Each data point represents a task, whose x-axis is the Relative Change (left) and Spearman's rank correlation coefficient (right) of $\lambda$ between the two models and the y-axis is the Relative Change or Spearman's rank correlation coefficient of $k$. We also suffix the task name with the unit granularity and the subset

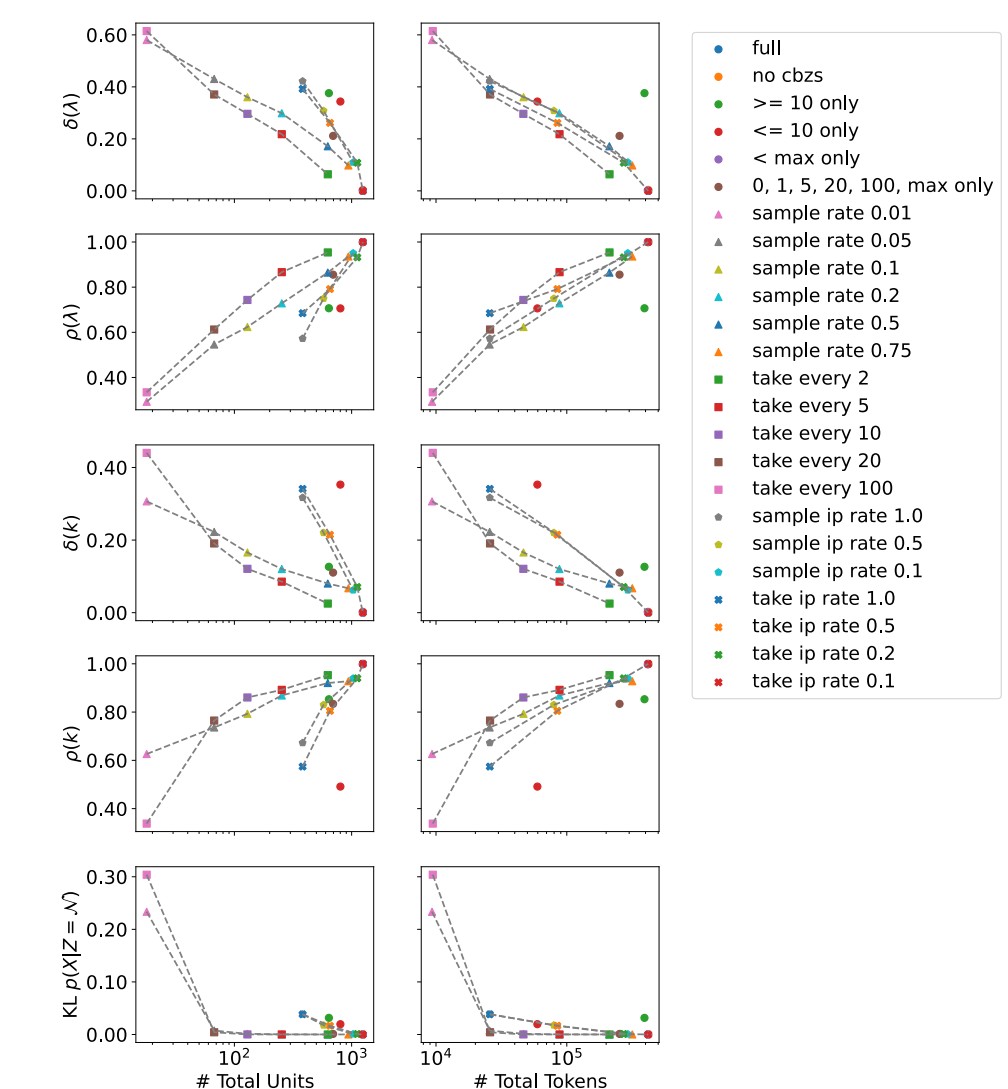

Figure 12: Relative change ($\delta$), Spearman's rank correlation coefficient ($\rho$) of $\lambda$ and $k$ and the KL divergence of $p(X|Z = \mathcal{N})$ between sampling strategies for the L-Eval TOEFL task. These strategies include sampling at a fixed rate $r$ (**sample rate**), taking every $N$-th (fixed) candidate in the sequence (**take every**), sampling inversely proportionally to the observation length (i.e. sampling more for short observations and less for long observations, **sample ip rate**), and taking every $N$-th (inversely proportionally to the observation length) candidate (**take ip rate**), as well as observing only long observations (**>= 10 only**), short observations (**<= 10 only**), and wider observation length intervals (**0, 1, 5, 20, 100, max only**). Results using the same strategy series are connected with dashed lines. We use the number of total units and the number of total tokens as the x-axes.

(COW or PIG) if any. The median $\delta(\lambda)$ and $\delta(k)$ are 0.30 and 0.16, and the median $\rho(\lambda)$ and $\rho(k)$ are 0.43 and 0.41, which fall into the moderate correlation category.

There are several reasons for the disagreement between the two models. First, similar to our investigation for unit granularities in Appendix K.2, we found there are a large number of cases where, despite a small $\rho$, the corresponding $\delta$ is also close to 0, especially $\rho(k)$ and $\delta(k)$. Examples include L-Eval OpenReview, LongBench MuSiQue's PIG subset (using "b" unit), LongBench HotpotQA's PIG subset (using "o" or "b" unit), etc. This happens when $k$ takes mostly 1 but also a few other small values.

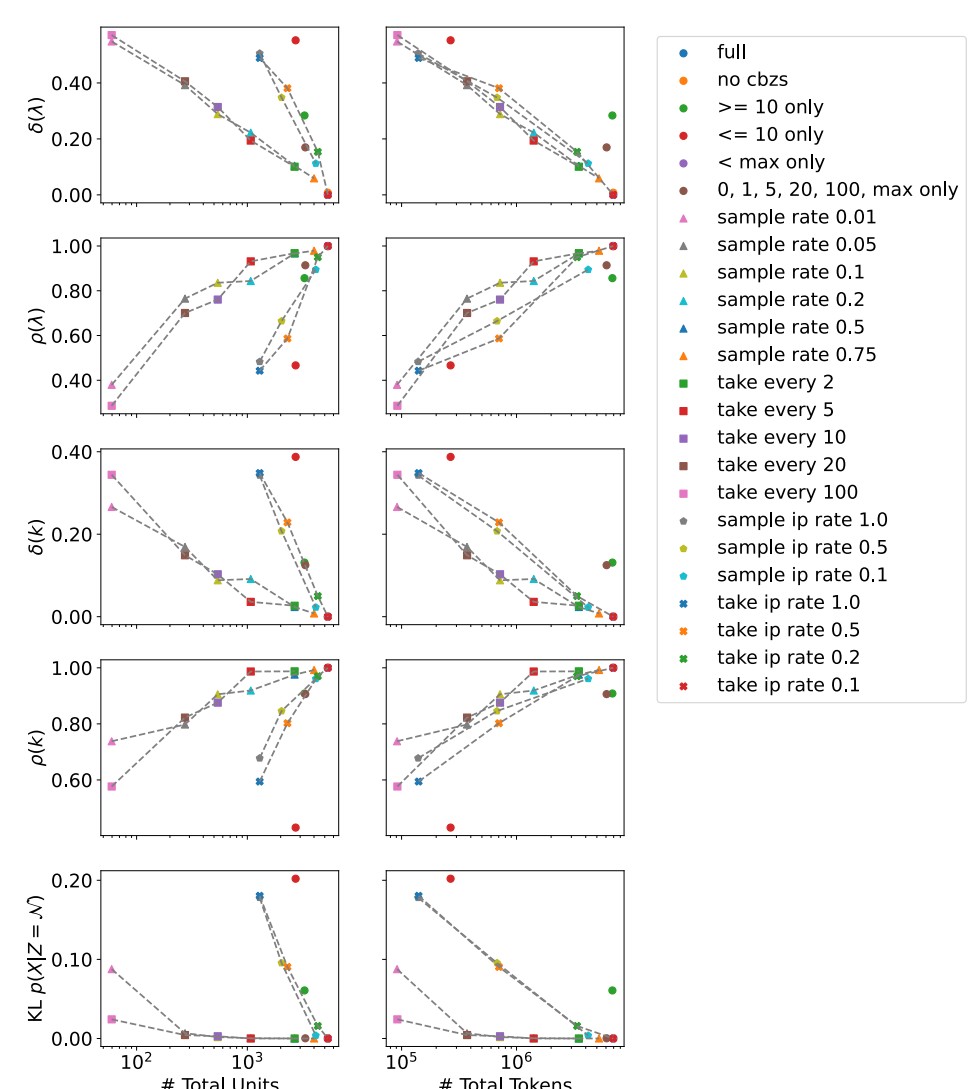

Figure 13: Relative change ($\delta$), Spearman's rank correlation coefficient ($\rho$) of $\lambda$ and $k$ and the KL divergence of $p(X|Z = \mathcal{N})$ between sampling strategies for the L-Eval SFcition task. These strategies include sampling at a fixed rate $r$ (**sample rate**), taking every $N$-th (fixed) candidate in the sequence (**take every**), sampling inversely proportionally to the observation length (i.e. sampling more for short observations and less for long observations, **sample ip rate**), and taking every $N$-th (inversely proportionally to the observation length) candidate (**take ip rate**), as well as observing only long observations (**>= 10 only**), short observations (**<= 10 only**), and wider observation length intervals (**0, 1, 5, 20, 100, max only**). Results using the same strategy series are connected with dashed lines. We use the number of total units and the number of total tokens as the x-axes.

Next, the model's short or median-length context understanding capability also matters. For example, in the LongBench PassageRetrieval task or the L-Eval LongFQA task, we found the PaLM 2-S model tends to mispredict (i.e. high $P(x = 0)$) and the Gemini 1.5 Flash model more likely predicts correctly or refuses to predict (i.e. high $P(x = 1)$ or $P(x = \emptyset)$). We manually look into the outputs from both models for the most representative holistic understanding focused problem (the 106th line) in the LongBench PassageRetrieval task. We found that the PaLM 2-S model tends to completely ignore the instruction that "the answer format must be like 'Paragraph 1', 'Paragraph 2', etc.", and outputs the paragraph number directly instead. The Gemini 1.5 Flash model follows the instruction

Table 8: Relative change ($\delta$) and Spearman's rank correlation coefficient ($\rho$) of $\lambda$ and $k$ and KL $P(x|z = \mathcal{N})$ between different unit granularity selections for the same tasks and subsets.

| TASK | REF | TEST | $\delta(\lambda)$ | $\rho(\lambda)$ | $\delta(k)$ | $\rho(k)$ | KL |
|------|-----|------|------|------|------|------|-----|
| | | COW Scenario | | | | | |
| L-Eval CodeU | l | b | .80 | .82 | .00 | NaN | .02 |
| HotpotQA | b | o | .57 | .55 | .24 | .56 | .18 |
| 2WikiMultiHopQA | b | o | .54 | .58 | .21 | .50 | .04 |
| MuSiQue | b | o | .67 | .64 | .20 | .51 | .11 |
| | | PIG Scenario | | | | | |
| HotpotQA | b | o | .39 | .47 | .08 | .73 | .04 |
| 2WikiMultiHopQA | b | o | .35 | .84 | .06 | NaN | .07 |
| MuSiQue | b | o | .75 | .80 | .00 | NaN | .02 |
| BAMBOO PrivateEval 4K | l | b | .67 | .78 | .60 | .77 | .02 |
| BAMBOO PrivateEval 16K | l | b | .55 | .71 | .67 | .64 | .00 |
| LongBench MultiNews | ls | o | .91 | .35 | .00 | NaN | .09 |

much better although sometimes formats using a markdown syntax e.g. "**Paragraph 3**". These answers are considered incorrect by the accuracy metric. Although we can manually tune the prompt and/or the postprocessing steps to normalize the outputs and the targets, we try to keep the process simple, since we believe the probing models' mistakes are inevitable nevertheless.

Also, we found that pre-existing internal knowledge learned during training can be another reason for the disagreement, although we have filtered out the problems as long as they are labeled as **Category I** with either model. L-Eval TOEFL is an example. With PaLM 2-S model, 15% of the problems are labeled as **Category I** and 24% are labeled as **Category II**, compared to 1% and 7% with Gemini 1.5 Flash model. Although we explicitly filtered the **Category I** problems, the whole distribution also tends to shift from **Category II** to **IV** or **V**.

We plot the parameters $\lambda$ and $k$ estimated for the three tasks: LongBench PassageRetrieval, L-Eval LongFQA, and L-Eval TOEFL, using Gemini 1.5 Flash and PaLM 2-S models in Figure 15. We also discuss the probing model selection criteria further in Appendix L.

### K.4 BINARIZATION THRESHOLD IN ADAPTING COW ASSUMPTION FOR CONTINUOUS SCORES

In this paper, we propose to use Hartigans' Dip Test to first identify the problems that have been scored using a continuous metric, such as F-1, ROUGE, and EditSim, and then binarize the scores using a predefined threshold before apply the COW assumption. Based on the observation from Figure 6, we believe 0.5 (or 50 percentage) is a reasonable threshold for these tasks. In fact, this is a subjective decision. The most reasonable threshold should depend on the task. Tasks that require to output longer (or shorter) texts may expect a lower (or higher) threshold.

We first, in Figure 16, plot the category assignments with each threshold selection for all the COW only tasks as well as COW subsets. In the latter case, we also compare with the category assignments of their PIG subsets, used as references only, since they are estimated from completely disjoint sets. We found that there are a few common patterns. The category assignment for BAMBOO SenHallu 4K/16K and AbsHallu 4K/16K, LongBench TriviaQA, SamSum, or RepoBench-p barely changes as the threshold changes, suggesting that their score distributions have two modes near 0 and 1. For all other tasks, we see that, as we increase the threshold, fewer **Category II** and/or **III** labels and more **Category V** and/or **IV** labels are assigned. In these cases, although each COW problem tends to have binary scores, as identified by the Hartigans' Dip Test, its two modes and the expected threshold differs across the problems. We may need to consider to use a per-problem threshold in a future work.

Then, in Figure 17, we plot the Relative Change ($\delta$) and Spearman's rank correlation coefficient ($\rho$) of $\lambda$ and $k$ between using a threshold of 50 and using thresholds of 0, 0.25, 0.75, 1. Each data point represents a task, whose x-axis is $\delta(\lambda)$ or $\rho(\lambda)$ between the assumptions and the y-axis is $\delta(k)$

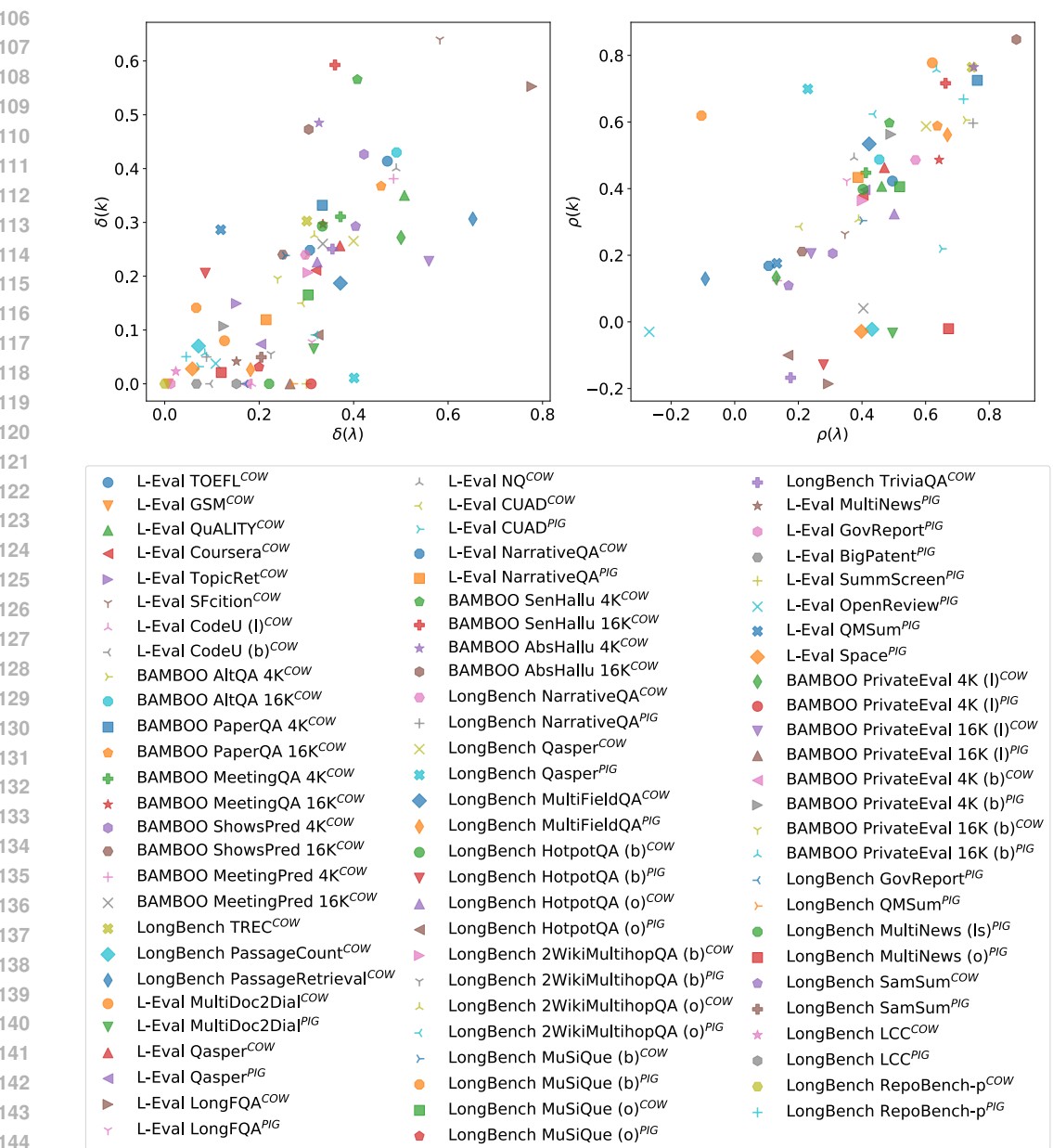

Figure 14: Relative Change ($\delta$) and Spearman's rank correlation coefficient ($\rho$) of $\lambda$ and $k$ between the Gemini 1.5 Flash model and the PaLM 2-S model across the tasks. Each data point represents a task, whose x-axis is the Relative Change ($\delta$) or Spearman's rank correlation coefficient ($\rho$) of $\lambda$ between the two models and the y-axis is the Relative Change ($\delta$) or Spearman's rank correlation coefficient ($\rho$) of $k$. We also suffix the task name with the subset (COW or PIG) and the unit granularity if any.

or $\rho(k)$. We found that $\delta(\lambda)$ and $\delta(k)$ are mostly below 0.48 and 0.37 across the tasks (except one task with each threshold) and $\rho(\lambda)$ and $\rho(k)$ are above 0.48 and 0.41 (except two tasks with each threshold), when the threshold is changed from 0.5 to either 0.25, 0.75, or even 1 across the tasks. When the threshold is changed to 0, we see much large $\delta(\lambda)$ and $\delta(k)$ and small $\rho(\lambda)$ and $\rho(k)$, suggesting the threshold must be greater than 0.

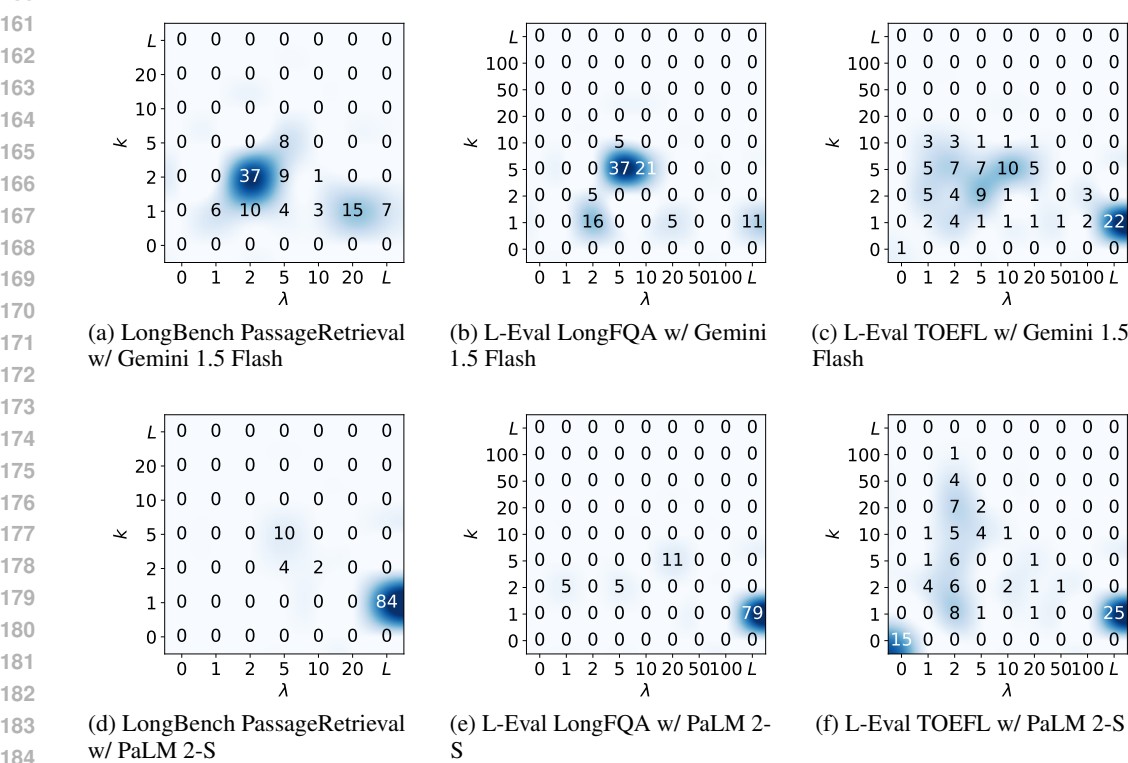

Figure 15: Parameters $\lambda$ and $k$ estimated using Gemini 1.5 Flash and PaLM 2-S models.

### K.5 Same Tasks From Different Benchmark Suites

We hypothesize that the different category distribution between the task versions in two benchmark suites can be explained by their distinct problem selection schemes. We provide the median context lengths for the four tasks used in both benchmark suites in Table 9. We see that the two versions of the Qasper task have very similar median context lengths, resulting in similar category assignments. L-Eval versions of the MultiNews task and the NarrativeQA task have more holistic understanding "flavor" than the LongBench versions, where we found the L-Eval versions often have longer contexts. In contrast, the L-Eval version of the GovReport task has fewer holistic understanding problems than the LongBench version, where the L-Eval version has much shorter median context length compared with the LongBench version.

Table 9: Median context length of problems (measured in tokens) in the four tasks that exist in both benchmark suites.

|  | Qasper | MultiNews | NarrativeQA | GovReport |
|---|---|---|---|---|
| L-Eval | 4,725 | 3,851 | 2,376 | 4,649 |
| LongBench | 4,791 | 2,150 | 1,373 | 8,955 |

### K.6 Application In Model Development: KV Cache Update Schedule

As detailed in Section 1, this work is motivated by our and others' observations that different efficient long context LLM architectures may behave differently for different categories of long context tasks, namely retrieval focused and holistic understanding focused. In this section, we present a case study by observing the benefit of the least recently attended (LRA) schedule (Yang & Hua, 2024) for different long context tasks.

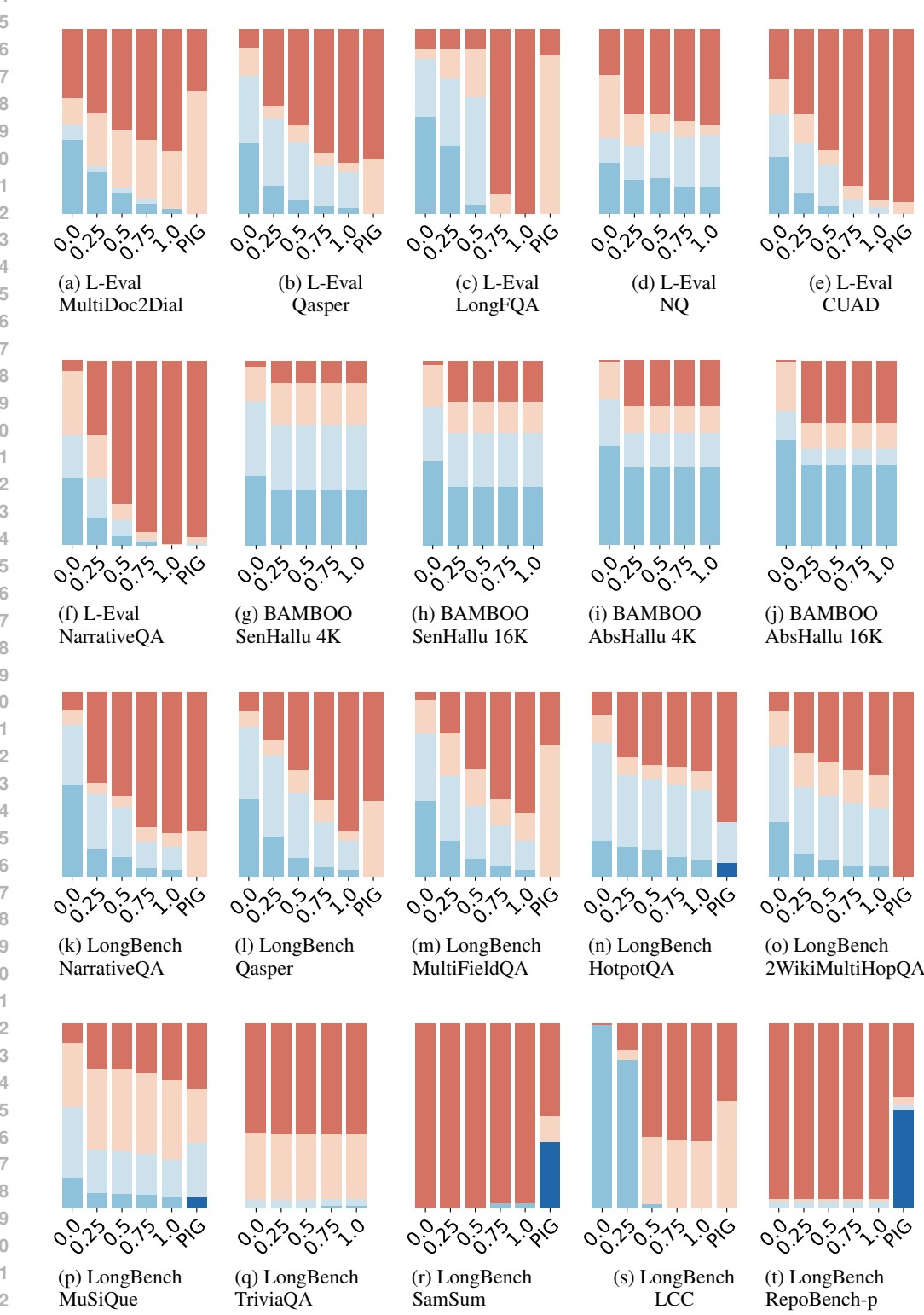

Figure 16: Category assignments with each threshold selection for all the COW only tasks as well as COW subsets. In the latter case, we also compare with the category assignments of their PIG subsets as reference.

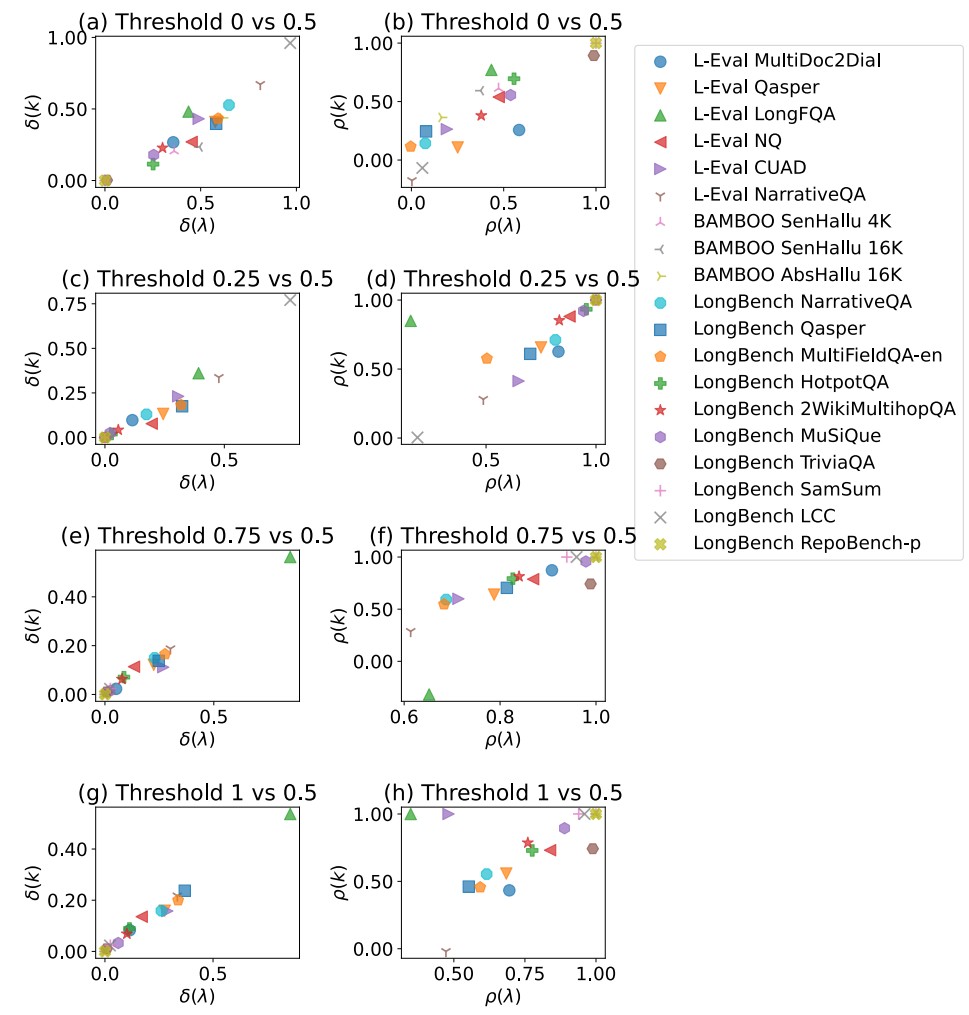

Figure 17: Relative Change ($\delta$) and Spearman's rank correlation coefficient ($\rho$) of $\lambda$ and $k$ between using a threshold of 0.5 and using thresholds of 0, 0.25, 0.75, 1 across all COW tasks or subsets. Each data point represents a task, whose x-axis is $\delta(\lambda)$ or $\rho(\lambda)$ between the assumptions and the y-axis is $\delta(k)$ or $\rho(k)$.

In a nutshell, it splits the long input into $C$ chunks, each having a length of $S$ and processes one chunk at each step. The LRA updates the KV cache by keeping only the top $N$ attended KVs at each step and dropping the other KVs. One advantage is that it can take an arbitrarily long input sequence without a set limit. Also, it reduces the computation complexity by a factor of $C$, compared to the vanilla attention taking the full $CS$ sequence as a whole. Like many cache update schedules, these efficiency benefits come at a cost: the least attended KV at certain stage may be essential for future steps but have to be removed from the cache, potentially hurting the long context task performance.

Intuitively, a retrieval focused task often requires a relatively small KV cache to store only the most relevant short span, **only if the task instruction or question is prefixed to the context**. This should translate to a phenomenon that by reducing the size of the chunk and/or cache capacity, we should see a retrieval focused task performance mildly drops. In contrast, when **the task instruction is suffixed to the context**, the attention score reflects the importance of each token in the "text generation task" in a query-independent fashion, and therefore the cache is not well maintained to properly answer the question. We expect that the performance should drop more in this case. A holistic understanding task, on the other hand, may expect a large KV cache. When we reduce the size of the KV cache, the performance should generally drop more. Moreover, putting the question or task instruction before or

after the context should not matter. When the question is suffixed to the context, the cache should have already compressed itself to keep only the essential information from the entire context.

In the preliminary experiments, we evaluate the LRA schedule using four tasks: two (L-Eval TOEFL and BAMBOO MeetingQA 4K) from the retrieval focused task categories, and two (LongBench NarrativeQA and L-Eval Qasper) from the holistic understanding focused task category. We use $S = N = 4096$ (large KV cache) and $S = N = 1024$ (small KV cache) respectively and report the relative performance percentage between the two cache sizes in Table 10.

Table 10: Relative performance percentage (the evaluation score using $S = N = 1024$ divided by the evaluation score using $S = N = 4096$).

|  | TOEFL | MeetingQA 4K | NarrativeQA | Qasper |
|---|---|---|---|---|
| Question is prefixed | 91.03% | 93.33% | 74.46% | 66.78% |
| Question is suffixed | 70.09% | 86.84% | 85.06% | 68.59% |

We see that the results align perfectly with our intuition and are consistent between the tasks within the same focus category. This suggests that if we want to utilize LRA to improve the efficiency of a long context application, we need to understand the focus category. If it is a retrieval focused task, we would encourage to state the question or task instruction before the context. If it is a holistic understanding focused task, we may suggest to delay the description of the instruction until the context is fully presented, which often gives better results.

## L  LIMITATIONS

First, we list the limitations of our proposed DOLCE framework.

**Mixture assumption.** The background noise module and the oracle module do not have to be combined using our "additive" mixture assumption. In fact, we also implement an alternative "multiplicative" or generative assumption. Under the multiplicative assumption, we continue to use the random variable x to denote the observed outcome, but we use the random variable z to denote the latent oracle outcome, which can take the same values as x. Then, $p(\mathrm{x}|\mathrm{z})$ represents the "transition" between the oracle and the actual outcome. We found that the parameters are more difficult to estimate if not regularized. Specifically, we sometimes learned a transition parameter $p(\mathrm{x} = 1|\mathrm{z} = 0)$ or $p(\mathrm{x} = 0|\mathrm{z} = 1)$ close to 1, which should rarely happen given that even the probing model should at least behave reasonably even when it is not confident.

**Oracle component assumptions.** We propose two assumptions: COW and PIG and derive $\pi$ and $\rho$ in Section 3, which may not be accurate for all the tasks. In fact, we are not restricted from using only the proposed assumptions. First, we can extend the COW assumption by further allowing noncontiguous ground-truth spans, or extend the PIG assumption to accommodate ground-truth aspect spans with length greater than one, which should allow us to further apply to synthetic needle-in-the-haystack task variants, e.g. FLenQA (Levy et al., 2024), BABILong (Kuratov et al., 2024). Next, we can also assume that we must find the aspects and put them in a certain order in order to answer the problem, which can help distinguish sequential reasoning tasks from summarization-like problems using "divide-and-conquer" (Levy et al., 2024). Finally, we may relax this restriction by allowing that any unit may be used by multiple ground truth spans. However, in the PIG scenario, the $\lambda$ ground-truth aspects are distributed independently of other ground-truth spans, and hence the probability becomes a function of only $k$ but not $\lambda$ any more. If we continue to model $\lambda$ under this additional "overlappable" aspect assumption, we need to make further assumption that the probability $\rho(s, \lambda, k; L, C)$ follows another distribution, where $\lambda$ can still be relevant to all the moments except the mean. We leave this for future work.

**MLE objective & sampling strategy.** We use MLE as the objective as it seems the most straight-forward formulation of the problem. But it does not mean this is the only objective. Also, we may modify the MLE formulation. For example, we currently give each sample an equal weight in our experiment. Alternatively, we can also give each observation length an equal weight in the likelihood function, regardless of the number of samples obtained using this observation length. As we show in Appendix K.1, small sample sizes often hurt the estimation accuracy for $\lambda$ and $k$ when the heuristic

sampling strategies are used. We may also improve the sampling strategy, i.e. by using a dynamic strategy that can shift to the next start position based on the previous outcome(s), or one that uses some global heuristics similar to important sampling. We can also combine retrieval methods into the sampling process to initialize the "importance" scores.

**Probing model.** Although our framework is designed to tolerate observation noises from using a probing model, its parameter inference effectiveness may still be impacted by the probing model, especially when it ignores our instructions. For example, if the probing model decides not to refer to the provided context, and instead retrieves the answer directly from its internal knowledge, it essentially wastes this problem, which may eventually be labeled as **Category I** or **II**. Another example is when the model has a conceited or humble characteristic by underusing or overusing "IDK" when it is not confident to answer. Although the framework can learn the latent tendency of the background noise component from the collective outcomes across the problems, the parameters of each problem are eventually determined based on the probing model's outcomes relative to the peer problems. If all the evaluation outcomes are "0", the mixture model assumption could hardly distinguish the source between the background noise component or the oracle component. In this work, we chose to use mid-sized models that should be capable of understanding short context texts and following instructions. We intentionally avoided using larger models due to their strong closed-book and zero-shot capabilities as a result of memorization of knowledge.

**Evaluation.** Evaluation plays a very crucial role in this process. While objective questions evaluated using accuracy as the metric are generally reliable, subjective and generative questions evaluated using ROUGE can be problematic, since the noise from the ROUGE scores can exceed the "denoising" allowance of our framework. In our work, we do not use any external script executor, human or LLM-as-a-rater service in this process, although we believe they should help improve the accuracy of the inferred parameters.

Besides, we note that our paper has other limitations, include

**Human evaluation verification.** We did not employ human annotators to confirm the estimated $\lambda$ and $k$. We only manually checked the most representative examples for several tasks beyond QuALITY and LongFQA. We note that more recent benchmarks, e.g. Karpinska et al. (2024) and Wang et al. (2024c), have started to manually label the difficulty categories using their own taxonomy. We can also consider to compare our automatically labeled categories with their manual difficulty categories.

