# OpenReview forum: "Retrieval Or Holistic Understanding? Dolce: Differentiate Our Long Context Evaluation Tasks"
_ICLR.cc/2025/Conference — Submitted to ICLR 2025_

### Official Review · Reviewer_5Yg3 · 2024-10-25

**Soundness:** 2
**Presentation:** 3
**Contribution:** 2
**Rating:** 5
**Confidence:** 2

**Summary:**

This paper proposes DOLCE, a framework that parameterizes the complexity $\lambda$ and the redundancy $k$ of long context understanding tasks and categorizes the tasks based on these values. For a given task, DOLCE estimates the complexity $\lambda$ as the minimum length of context required to correctly answer the task, based on the correct response rate of LLMs vs. the input context length (number of consecutive units). Since the correct response rate is noisy, $\lambda$ is estimated by dedicated mixture models. The paper reports $\lambda$ and $k$ estimated by DOLCE for 44 tasks in three recent benchmarks (L-Eval, BAMBOO, and LongBench) as well as the categorization results of them.

**Strengths:**

- The motivation is convincing: improving the LLMs' capability of long context understanding is a key challenge. It can be useful to categorize the types of problems so that improvements can be considered separately for each type of problem.

- It is reasonable to design the proposed model to explicitly incorporate the possibility that LLMs may give correct answers by chance (background noise component) and that even oracle models may give "correct" wrong answers in a given context.

**Weaknesses:**

W1. The proposed model DOLCE assumes that the minimum number of "consecutive" units (typically sentences) required to correctly answer a question determines the complexity of the question \lambda. However, it seems to me that it is not always necessary to look at consecutive units in order to properly understand the context (there may be irrelevant sentences in between relevant sentences to understanding the context). This assumption appears to be somewhat unreasonable.


W2. As the authors acknowledge, the category mapping given in Fig. 1 is quite subjective and has no objective basis for its validity.


W3. This paper reports a number of estimated results of the complexity $\lambda$ and the redundancy $k$ for various problems as well as the categorization results of the problems based on these values. However, whether these estimation results correctly evaluate the complexity and redundancy of the problems is largely dependent on the qualitative observations on a small number of examples, such as Table 2, and lacks an objective basis. It is understandable that quantitative evaluation is rather difficult, but as discussed in L. Limitation, evaluation by human evaluation verification would be possible. These factors somewhat obscure the validity of DOLCE.


W4. As explained in the introduction section, one of the motivations for DOLCE is to improve LLMs' ability for long context understanding. However, whether it is actually useful for this goal has not been evaluated and is not clear.


W5. Some minor points:

L178: It would be good to describe the meaning of subscriptions i and j here. As given later in L236, they sould be j-th observation for the i-th problem.

L190: The authors claim that the background noise component is modeled by a "nonparametric" model, but I would like an explanation of what they mean by "nonparametric" here.

**Questions:**

The biggest problem with this paper would be the lack of objective evidences for the validity of DOLCE for the results of task complexity estimation and categorization and for its usefulness (W2-W4). I would appreciate it if the authors could provide supporting evidences.

---

> ### Author Response · Authors · 2024-11-29
> **(1/2) Appendix K.6 added, hoping to show case how task categorization may help long context capability understanding and architecture design**
>
> Thanks for your comments and suggestions!
>
> I uploaded a newer version, where the major change, as per your request in W4 and the question section, is to include **an additional appendix section (K.6)** to show our preliminary results of a case study that utilizes the task categories to better understand how much an efficient KV cache update policy proposed in an earlier work may hurt task performance, where we observed two distinct behavioral patterns between retrieval focused tasks and holistic understanding focused tasks. I hope this can help understand **how task categorization may help long context capability understanding and architecture design**. Really appreciate if you can take another look, and I'll also summarize the key messages in the following answer to W4.
>
> Now, let me also try to address your questions and concerns.
>
> **Re W1**: Your understanding that _our COW scenario assumes that a single ground-truth span must be consecutive_ is mostly correct. I also believe that we could get more accurate estimation if we relax this assumption, as we also mentioned in the Limitations section (Appendix L), where we suggested to "extend the COW assumption by further allowing noncontiguous ground-truth spans" (Line 2359 in the newer version). In fact, you may find that there can be multiple ways to further extend the assumptions in the "oracle component assumptions" subsection.
>
> In this paper, we keep this rather simpler assumption for the following reasons:
>
> 1. **Our PIG scenario does support nonconsecutive ground-truth spans**. However, it replaces this constraint with another constraint that the length of a single aspect span must be one (i.e. one single sentence is an aspect). We might instead choose to use the PIG scenario if we feel the task fits it better.
>
> 2. We want to make sure that both the COW and PIG assumptions must have only two variables (namely \lambda and k), which **simplifies greatly our derivations of the probabilistic functions and the estimation of the parameters**. If we further allow to support nonconsecutive non-unit spans for the COW scenario (or other ways of extensions), we need to have more variables.
>
> 3. We found that **most real world long context inputs in our current COW task sets often have continuity**, i.e. it is less likely that there exists a large portion of irrelevant content inside a ground-truth span. (However, it is quite common in PIG tasks.) Our studies on the sampling strategies (Appendix K.1) indirectly confirms the existence of continuity in the sequence, i.e. the correctness of a span is positively correlated with that of its neighboring span (Line 1755). However, we do acknowledge there might be other tasks that do not follow this pattern. Most notable examples are various synthetic needle-in-the-haystack tasks.
>
> **Re W2**: Yes, the choice of the threshold parameters is subjective. However, we must note that it's equally subjective to decide if a given length (say 4K) is short or long, which depends on the infrastructure (edge device vs server), the use case (text only or multimodal), the training data (aka expected maximum input length seen at inference time), etc. In our work, we consider them as hyperparameters, and therefore, we **the developers can decide how long (\lambda_p and \lambda_q) and how many (k_p) should be considered too long or too many**. In our experiments, we use 1/3 and 2/3 of the maximum allowed context length (which is predetermined by on our infrastructure) as \lambda_p and \lambda_q across the tasks, making the result a fair comparison. Also please note that instead of the raw focus category distribution of each task (e.g. 10% Category III + 20% Category IV + 30% Category V), our work wants to emphasize the diversity in the distributions (i.e. some tasks have more Category V problems than others) and present the relative rankings of the tasks, where **the hyperparameters (\lambda_p, \lambda_q and k_p) play a much smaller role**.
>
> **Re W3**: I completely agree that a large-scale human assessment of the predicted categories should be much preferred. As you mentioned, we might want to prioritize the automatic approaches (Line 2403), followed by the costly manual annotation.

---

> > ### Author Response · Authors · 2024-11-29
> > **(2/2) Appendix K.6 added, hoping to show case how task categorization may help long context capability understanding and architecture design**
> >
> > **Re W4**: We added an appendix section K.6 to describe a pilot case study, where we studied a previously published work that proposed to improve the KV cache efficiency by replacing the vanilla FIFO policy with a LRA policy, where only the most attended K/V positions are kept in the KV cache. The original work suggested that by using LRA the task performance drops much less than using FIFO when we reduce the size of the KV cache (i.e. computation and storage complexity). We then observed the performance drops using four tasks: two (L-Eval TOEFLand BAMBOO MeetingQA 4K) from the retrieval focused task categories, and two (LongBenchNarrativeQA and L-Eval Qasper) from the holistic understanding focused task category. We found completely different behavioral patterns: a retrieval focused task performance mildly drops when the task instruction or question is prefixed to the context, and drops more when the task instruction is suffixed to the context. On the other hand, a holistic understanding task performance generally drops more. However, putting the question or task instruction before or after the context does not matter. This suggests that **if we want to utilize LRA to improve the efficiency of along context application, we need to understand the focus category**. If it is a retrieval focused task, we would encourage to state the question or task instruction before the context. If it is a holistic understanding focused task, we may suggest to delay the description of the instruction until the context is fully presented, which often gives better results. **We can hardly find these results if we do not have the focus categories of the tasks.**
> >
> > **Re W5 / L178**: Thanks for pointing it out! I've updated the submission. Yes, the variables should be explained at their first appearance.
> >
> > **Re W5 / L190**: Sorry for the confusion here. The meaning of "nonparametric" in our work is more pronounced in the oracle component, when compared with the parametric oracle component. By "parametric", we mean the probability that some outcome is observed follows a hypothetic probability function parameterized by \lambda and k (e.g. \pi(\lambda, k; L, C) in Appendix A.1). In contrast, we make no assumption about the the nonparametric oracle (or noise) component, so we have to "fall back" to the frequentist method by estimating the raw frequency of each outcome. We borrowed these two terms from the Bayesian (frequentist) statistics, where the semantics are not entirely identical to ours but quite similar.
> >
> > Thanks for the comments again! Please let us know if you find the additional appendix section (K.6) helpful and/or if you have further questions or concerns. I'm happy to address them.

---

> > > ### Comment · Reviewer_5Yg3 · 2024-12-02
> > >
> > > I would like to thank the authors for providing detailed responses and additional analysis. I now have a better understanding of the paper.
> > >
> > > W2: I agree with most of the authors' comments. I also agree that it is not easy to construct a completely nonsubjective categorization. However, the goal of this paper should be to characterize the problem with reference to the capabilities of current LLMs, i.e., machine learning models. If this is correct, I would think that the infrastructure and other conditions are secondary and that we should stick to a classification based purely on the LLM's capabilities and try to ensure some objective basis.
> > >
> > > W4: I would like to thank the authors for presenting new preliminary experimental results. The results are convincing to some extent. The improvements made in this preliminary experiment are possible under the assumption that the category of the problem is known. In practical applications, it is not always known in advance, so the process of identifying the category of the input task must be necessary. I would say further discussions may be necessary on this point.

---

### Official Review · Reviewer_aUvp · 2024-10-30

**Soundness:** 3
**Presentation:** 3
**Contribution:** 3
**Rating:** 6
**Confidence:** 1

**Summary:**

The paper presents the DOLCE framework which aims to automatically identify retrieval and holistic understanding focused problems from benchmark suites.

Proposes to sample short contexts from the full context and estimate the probability an LLM solves the problem using the sampled spans. Introduces a mixture model of a non-parametric background noise component and a parametric/non-parametric hybrid oracle component to find λ and k.

**Strengths:**

The idea of using a mixture model of background noise and oracle components to estimate these parameters for different scenarios (COW and PIG) is creative.
The theoretical foundation of the DOLCE framework seems solid. The derivations of probability functions for the COW and PIG scenarios based on the assumptions of the mixture model are detailed and mathematically sound.
The experimental setup is comprehensive, using 44 tasks from three benchmark suites.

**Weaknesses:**

1. The background noise module and the oracle module are combined using an “additive” mixture assumption. However, alternative assumptions such as “multiplicative” or generative assumptions could be explored further.

2. Using MLE as the sole objective might limit the framework’s performance. Alternative formulations of the MLE could be explored, such as giving different weights to samples or observation lengths.

3. The assumptions made in the DOLCE framework, such as the specific forms of the background noise and oracle components, may limit its generalizability to a wide variety of real-world language tasks. For example, in real-world scenarios, the distribution of relevant information within a long context may not follow the assumed patterns precisely. This could lead to inaccurate categorization of tasks as retrieval or holistic understanding focused.

4. The paper focuses on complexity in terms of span length (λ) and redundancy (k), but it may not fully capture the semantic complexity variation within long contexts. Different semantic relationships and levels of abstraction can significantly impact the difficulty of a task, yet these aspects are not comprehensively incorporated into the framework.

**Questions:**

see Weaknesses.

---

> ### Author Response · Authors · 2024-11-29
>
> Thanks for your comments and suggestions! Weaknesses 1, 2, 3 are related to our statements in the Appendix L (limitations) section. In particular, we have acknowledged in this section that
>
> 1. "The background noise module and the oracle module do not have to be combined using our "additive" mixture assumption. In fact, we also implement an alternative "multiplicative" or generative assumption." (Line 2348)
>
> 2. "We use MLE as the objective as it seems the most straight-forward formulation of the problem. But it does not mean this is the only objective. Also, we may modify the MLE formulation. For example, we currently give each sample an equal weight in our experiment. Alternatively, we can also give each observation length an equal weight in the likelihood function, regardless of the number of samples obtained using this observation length." (Line 2371)
>
> 3. "We propose two assumptions: COW and PIG and derive π and ρ in Section 3, which may not be accurate for all the tasks. In fact, we are not restricted from using only the proposed assumptions. First, we can extend the COW assumption by further allowing noncontiguous ground-truth spans, or extend the PIG assumption to accommodate ground-truth aspect spans with length greater than one..." (Line 2357)
>
> We believe **there are many future directions to study the alternative approaches in all aspects**. However, we have already had **(more than) enough content to put in a single submission and we are eager to share with the research community** before delving deeper. Re 3, in particular, our work proposed two assumptions (PIC and COW), due to the simplicity (two variables each), but our DOLCE framework is general and welcomes all alternative oracle assumptions. The MLE / EM based algorithm does not need to change.
>
> Re 4: I agree that the difficulty of a task is not necessarily related to the length or redundancy. For example, some tasks require longer ground-truth spans, but the texts are simple English, whereas other tasks require shorter ground-truth span, but the texts are difficult math derivation. Our framework, specially the oracle component (parameterized by \lambda and k), does not incorporate this. A few more comments on this:
>
> 1. It is quite difficult, as you suggested, to capture all these many factors, as the first step. We may revisit this in the future.
>
> 2. Our work originates from the long context understanding community, as well as a related RAG community. As a result, we mostly care about the length and the redundancy. The length is the only factor in the long context understanding. Other factors (like semantic difficulty) do exist in both short and long context understanding problems.
>
> 3. When the semantic difficulty is increased (without changing the length or redundancy), the probing model can "feel the pain". It may react in two different ways: (1) it can still handle it properly and succeed as if it is not difficult, or (2) it fails to handle it and returns a wrong answer. In the former case, all estimation remains accurate. In the latter case, the noise component comes into play. We expect the p(z = O) decreases, and p(z = N) increases and p(X | z = N) will help smooth the noise. However, we will expect the estimations of \lambda and k are less accurate, due to the decreased total membership p(z = O). At a high level, the nonparametric noise component takes all factors other than \lambda and k into its "naive" modeling. Estimation should be more accurate if we can model other factors explicitly like \lambda and k.

---

### Official Review · Reviewer_SwXU · 2024-10-31

**Soundness:** 2
**Presentation:** 2
**Contribution:** 2
**Rating:** 5
**Confidence:** 2

**Summary:**

The manuscript presents a novel framework, DOLCE, which aims to differentiate between retrieval-focused and holistic understanding-focused problems within long context evaluation tasks. The authors argue that understanding these distinctions is crucial for improving the capabilities of large language models (LLMs) in processing extensive textual data. The framework introduces a parameterization approach with λ (complexity) and k (redundancy) to categorize problems into predefined focus categories.

**Strengths:**

1. The DOLCE framework is innovative in its approach to categorizing long context evaluation tasks, providing a new lens through which to understand and improve LLMs.
2.  The paper employs a rigorous methodology, including a mixture model that combines non-parametric and parametric components, to estimate the parameters λ and k.
3. The authors provide a thorough empirical validation across 44 existing long context evaluation tasks, demonstrating the applicability and effectiveness of the DOLCE framework.

**Weaknesses:**

1.The main text of this paper has major flaws, it is difficult to understand the relationship between the author's theory and experiments, and the layout is confusing. In addition, retrieval and overall understanding have been mentioned in the larger model and are not innovative.
2. While the paper focuses on long context understanding, it would benefit from discussing the broader implications of the findings on other areas of natural language processing.
3. The authors should compare the DOLCE framework with existing methods for evaluating long context capabilities to highlight its advantages and potential limitations.
4. The paper could address the scalability and efficiency of the DOLCE framework, especially when applied to very large datasets or in real-time applications.
5. The manuscript could benefit from a deeper exploration into how retrieval and holistic understanding interact in the context of LLMs, potentially offering more nuanced insights.
6. A robustness analysis to evaluate how sensitive the framework's categorization is to variations in λ and k would strengthen the paper's contributions.

**Questions:**

Please refer to weakness.

---

> ### Author Response · Authors · 2024-11-29
>
> Thanks for your comments and suggestions!
>
> Now, let me also try to address your questions and concerns.
>
> **Re 1**: Maybe let me try to clarify the focus of our work. We do acknowledge that retrieval and overall understanding have been mentioned in prior work (as we summarized in Section 2). The reality is that retrieval tasks are usually branded as retrieval tasks (unchanged), but some people believe that long context understanding tasks must be more or less whole context understanding, which is not true. The motivations of our work include (1) we want to emphasize that both retrieval and holistic understanding tasks exist in the large body of long context understanding benchmarks, (2) they are two distinct task types, and the long context LLMs can behave quite differently on these two categories, so people should almost always report / group their LLM performance results on the retrieval subsets and holistic understanding subsets separately, (3) we propose a method to help automatically tells if a task / problem is retrieval or holistic understanding without any human intervention, which is novel.
>
> **Re 2**: Long context understanding itself is a very broad research topic in NLP nowadays. People have (1) proposed new capabilities under the long context understanding framework (i.e. many-shot in context learning, in context retrieval, etc.), (2) studied how to extend the context length of existing short-/mid-context LLMs with little continual pretraining, (3) studied efficient inference methods of long context models. Although this work focuses on NLP tasks, the proposed method can also be adapted to other modalities as well beyond natural language tasks.
>
> **Re 3**: I think you haven't quite understood the goal of the work. Please note that the DOLCE framework is NOT to evaluate long context capabilities. It is NOT a new long context benchmark (like those reviewed in Section 2). The goal of the DOLCE framework is to identify the focus category of an EXISTING benchmark task / problem. To the best of our knowledge, there's not any prior work that has tried to accomplish the same goal. We are the first work that can automatically identifies the focus category. Subsequent works can then compare their methods with our DOLCE framework.
>
> **Re 4**: For a given problem, the computation complexity of the Algorithm 1 / 2 is given by T|Λ||K||J|, where T is the iteration steps, |Λ| is the number of \lambda candidates, |K| is the number of k candidates, |J| is the number of samples. They all can be adjusted to fit the estimation budget if needed. However, reducing any of them may have a negative impact to the accuracy of the estimation. The complexity is also linear to the number of problems. So it makes no difference when applied to larger or smaller datasets, as long as the goal is to estimate for each single problem in the dataset. Real-time application? I think this is not a concept relevant to our work.
>
> **Re 5**: We added an appendix section K.6 to describe a pilot case study, where we studied a previously published work that proposed to improve the KV cache efficiency by replacing the vanilla FIFO policy with a LRA policy, where only the most attended K/V positions are kept in the KV cache. The original work suggested that by using LRA the task performance drops much less than using FIFO when we reduce the size of the KV cache (i.e. computation and storage complexity). We then observed the performance drops using four tasks: two (L-Eval TOEFLand BAMBOO MeetingQA 4K) from the retrieval focused task categories, and two (LongBenchNarrativeQA and L-Eval Qasper) from the holistic understanding focused task category. We found completely different behavioral patterns: a retrieval focused task performance mildly drops when the task instruction or question is prefixed to the context, and drops more when the task instruction is suffixed to the context. On the other hand, a holistic understanding task performance generally drops more. However, putting the question or task instruction before or after the context does not matter. This suggests that if we want to utilize LRA to improve the efficiency of along context application, we need to understand the focus category. If it is a retrieval focused task, we would encourage to state the question or task instruction before the context.  If it is a holistic understanding focused task, we may suggest to delay the description of the instruction until the context is fully presented, which often gives better results. We can hardly find these results if we do not have the focus categories of the tasks.
>
> **Re 6**: The \lambda and k are outputs of the system, not the inputs. So I cannot answer your concern regarding the framework's categorization if we vary \lambda and k. No, we can't vary \lambda and k. They are variables estimated by the system.

---

### Official Review · Reviewer_u9M1 · 2024-11-02

**Soundness:** 2
**Presentation:** 2
**Contribution:** 2
**Rating:** 6
**Confidence:** 3

**Summary:**

This paper introduces a framework for understanding and improving long-context capabilities in large language models (LLMs) by distinguishing between two key skills: retrieval and holistic understanding. The authors propose the DOLCE framework, which classifies tasks into focus categories based on complexity and redundancy parameters. Their methods can identify retrieval-focused and holistic-focused problems across 44 long-context evaluation tasks, classifying 0%-67% as retrieval and 0%-90% as holistic understanding-focused.

**Strengths:**

1. Long context understanding is crucial for enhancing LLMs' performance in real-world applications where retaining and processing extended information is essential.

2. To my knowledge, the framework’s ability to automatically categorize long-context tasks based on their focus—retrieval or holistic understanding—is novel.

**Weaknesses:**

1. The comparison on the right side of Figure 1 is confusing; it lacks clear explanations of the hyperparameters. Additionally, the description of 𝑘 for balanced and holistic understanding (marked as “any” in the right table) does not align with the illustration on the left side of Figure 1. It would be better to polish this figure.

2. The meaning of 𝑧 in Equation 1 is unclear and should be explicitly defined for better understanding. Please clarify it.

3. There are no theoretical results provided to guarantee the effectiveness of maximum likelihood estimation in this context. It would be great if the authors could provide any optimality analysis for this strategy.

4. The efficiency of observation sampling is not discussed, leaving uncertainty about its computational feasibility and scalability.

5. Lines 456-461 need further clarification regarding the differences between the various probing models and the reasons behind these differences. This would help readers understand the specific roles and implications of each model.

**Questions:**

Please refer to weaknesses.

---

> ### Comment · Reviewer_u9M1 · 2024-11-26
>
> The authors did not provide any rebuttal, so I kept my decision rating unchanged.

---

> ### Author Response · Authors · 2024-11-29
>
> Thanks for your comments and suggestions! With the updated submission, I added *an additional appendix section (K.6)** to show our preliminary results of a case study that utilizes the task categories to better understand how much an efficient KV cache update policy proposed in an earlier work may hurt task performance, where we observed two distinct behavioral patterns between retrieval focused tasks and holistic understanding focused tasks. I hope this can help understand how task categorization may help long context capability understanding and architecture design.
>
> Now, let me also try to address your questions and concerns.
>
> **Re 1**: Thanks for the suggestion! In the updated submission, I added the explanation of the three hyperparameters. I also updated "Any" to [1, L / \lambda_p] and [1, L / \lambda_q] respectively to make them clear. However, please note that they are actually equivalent to "Any", since we always have \lambda x k <= L (non-overlapping assumption), so we will never reach the upper-right triangle in the left figure (i.e. the upper-right white area above IV and V).
>
> **Re 2**: In the updated submission, I added that z is the latent variable, taking either the value O (for the oracle component) or N (for the noise component). Yes, the variables should be explained at their first appearance.
>
> **Re 3**: I completely agree that while the problem is formulated as a maximum likelihood estimation, the provided Algorithms 1 and 2 use EM in the outer loop and a brute-force based maximum in the inner loop. We haven't done any theoretical analysis in the work, while we only focus on the empirical side of the effectiveness. Thanks for the suggestion! We can study this further.
>
> **Re 4**: I have two interpretations of your concern: (1) you might wonder the feasibility of LLM inference (i.e. the process of sampling and observation of the outcomes), or (2) you might wonder, once the outcomes are collected, how the size of the samples impact the Algorithms 1 / 2 (i.e parameter estimation stage).
>
> First, (2) is generally more efficient than (1). The computation complexity of the algorithm is quite straightforward: for a given problem, it is T|Λ||K||J|, where T is the iteration steps, |Λ| is the number of \lambda candidates, |K| is the number of k candidates, |J| is the number of samples. The complexity is also linear to the number of problems. Moreover, the |Λ| and |K| loops can be run in parallel. So (2) should not be a concern.
>
> The actual bottleneck may be (1). There are also two factors: (a) the number of samples and (b) the length of each sample. The former is a linear factor and grows linearly to the full length of the entire context, whereas the latter is a linear or quadratic factor and determined by the choices of \lambda, which can be controlled (i.e. constant, if we only care about a fixed set of \lambda values, 1, 2, 5, 10, ...). So overall, the sampling complexity generally scales linearly to the full length of the entire context. It may be further improved by utilizing a retrieval method that ranks the spans, instead of "blind" sampling, to achieve sublinear scalability.
>
> **Re 5**: Thanks for the suggestion. The goal of the work is to treat the probing model as a black box, as long as it can somewhat reasonably produce answers for short-/mid-context inputs. We added this study, trying to address the concern that if I have multiple (black box) probing model candidates, how we should choose which one to use, and at a higher level, whether the choice of the probing model matters. The message that we would like to deliver here is that it does not matter too much.
>
> Then back to your great question, based on their respective technical reports, the two models are both Transformer based models, but they differ mostly in (1) training dataset mixture, (2) training procedure, (3) various tweaks of Transformer architecture -- pretty much every step of the training. The details and the reasons for these differences are hard to known, since they are third-party commercial models. However, this does not prevent us from using them and concluding that our DOLCE framework is somewhat robust to the probing model. On the other hand, our method can potentially reveal the differences by identifying the Categories I and II problems.

---

### Meta-Review · Area_Chair_xhT9 · 2024-12-11

**Metareview:**

This paper claims that there are two major distinct capabilities in long context understanding, namely retrieval and holistic understanding. Specifically, this paper presents the Dolce framework, which parameterizes each problem by complexity and redundancy and assigns to one of five predefined focus categories. To find the values of complexity and redundancy for each problem, the authors further propose a mixture model of a non-parametric background noise component and a parametric/non-parametric hybrid oracle component, where they derive the probability functions for both the correct-or-wrong (COW) scenario and the partial-point-in-grading (PIG) scenario. The proposed methods can identify 0% to 67% of the problems are retrieval focused and 0% to 90% of the problems are holistic understanding focused across 44 existing long context evaluation tasks.

The topic of this paper is interesting, and the paper is also written in a clear way. The experiments well demonstrate the effectiveness of the proposed method.

**Additional Comments On Reviewer Discussion:**

This paper is potentially interesting. The AC organized a discussion on this paper among the reviewers. In the discussion, the reviewers mainly doubt about the practicality of this approach. Besides, more objective and quantatitive evaluations are also expected. Therefore, it would be better if this paper can be improved from these aspects.

---

### Decision · Program_Chairs · 2025-01-22

Reject